# Dataset Distillation using Neural Feature Regression

**Yongchao Zhou**
Department of Computer Science
University of Toronto
yongchao.zhou@mail.utoronto.ca

**Ehsan Nezhadarya**
Toronto AI Lab
LG Electronics Canada
ehsan.nezhadarya@lge.com

**Jimmy Ba**
Department of Computer Science
University of Toronto
jba@cs.toronto.edu

## Abstract

Dataset distillation aims to learn a small synthetic dataset that preserves most of the information from the original dataset. Dataset distillation can be formulated as a bi-level meta-learning problem where the outer loop optimizes the meta-dataset and the inner loop trains a model on the distilled data. Meta-gradient computation is one of the key challenges in this formulation, as differentiating through the inner loop learning procedure introduces significant computation and memory costs. In this paper, we address these challenges using neural Feature Regression with Pooling (FRePo), achieving the state-of-the-art performance with an order of magnitude less memory requirement and two orders of magnitude faster training than previous methods. The proposed algorithm is analogous to truncated backpropagation through time with a pool of models to alleviate various types of overfitting in dataset distillation. FRePo significantly outperforms the previous methods on CIFAR100, Tiny ImageNet, and ImageNet-1K. Furthermore, we show that high-quality distilled data can greatly improve various downstream applications, such as continual learning and membership inference defense. Please check out our webpage at https://sites.google.com/view/frepo.

## 1 Introduction

Knowledge distillation [1] is a technique in deep learning to compress knowledge for easy deployment. Most previous works focus on model distillation [2, 3] where the knowledge acquired by a large teacher model is transferred to a small student model. In contrast, dataset distillation [4, 5] aims to learn a small set of synthetic examples preserving most of the information from a large dataset such that a model trained on it can achieve similar test performance as one trained on the original dataset. Distilled data can accelerate model training and reduce the cost of storing and sharing a dataset. Moreover, its highly condensed and synthetic nature can also benefit various applications, such as continual learning [5–8], neural architecture search [5, 7], and privacy-preserving tasks [9, 10].

Dataset distillation was first studied by Maclaurin et al. [11] in the context of gradient-based hyperparameter optimization and subsequently Wang et al. [4] formally proposed dataset distillation as a new task. Dataset distillation can be naturally formulated as a bi-level meta-learning problem. The inner loop optimizes the model parameters on the distilled data (meta-parameters), while the outer loop refines the distilled data with meta-gradient updates.

One key challenge in dataset distillation is computing the meta-gradient. Several methods [4, 11–13] compute it by back-propagating through the unrolled computation graph, but they often suffer from

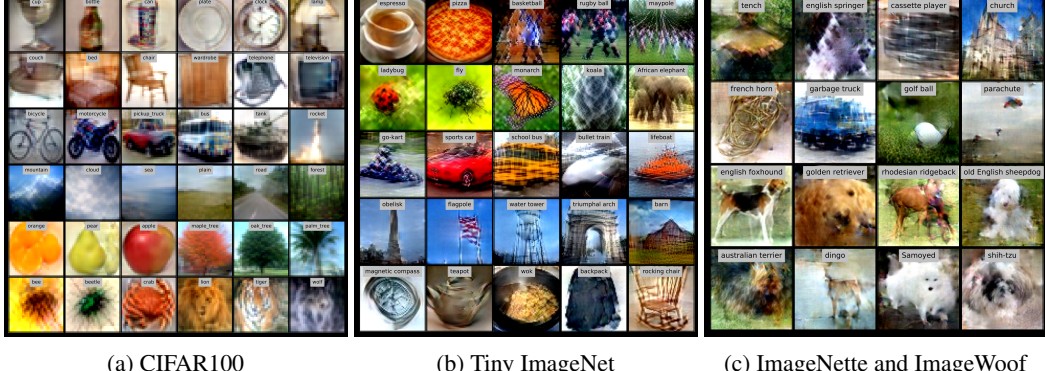

| (a) CIFAR100 | (b) Tiny ImageNet | (c) ImageNette and ImageWoof |

Figure 1: Example distilled images from 32x32 CIFAR100, 64x64 Tiny ImageNet, and 128x128 ImageNet Subset. The images look real and transfer well to different architectures. They can be used for various downstream applications, such as continual learning and membership inference defense.

huge compute and memory requirement [14], training instability [15, 16], and truncation bias [17]. To avoid unrolled optimization, surrogate objectives are used to derive the meta-gradient, such as gradient matching [5, 7, 18], feature alignment [8, 19], and training trajectory matching [20]. Nevertheless, a surrogate objective may introduce its own bias [19], and thus, may not accurately reflect the true objective. An alternative is using kernel methods, such as Neural Tangent Kernel (NTK) [21], to approximate the inner optimization [22, 23]. However, computing analytical NTK for modern neural network can be extremely expensive [22, 23].

Even with an accurate meta-gradient, dataset distillation still suffers from various types of overfitting. For instance, the distilled data can easily overfit to a particular learning algorithm [4, 13, 20], a certain stage of optimization [13, 19], or a certain network architecture [5, 7, 20, 22, 23]. Meanwhile, the model can also overfit the distilled data during training, which is the most common cause of overfitting when we train on a small dataset. All these kinds of overfitting impose difficulties on the training and general-purpose use of the distilled data.

We propose an efficient meta-gradient computation method and a "model pool" to address the overfitting problems. The bottleneck in meta-gradient computation arises due to the complexity of inner optimization, as we need to know how the inner parameters vary with the outer parameters [24]. However, the inner optimization can be pretty simple if we only train the last layer of a neural network to convergence while keeping the feature extractor fixed. In this case, computing the prediction on the real data using the model trained on the distilled data can be expressed as a kernel ridge regression (KRR) with respect to the conjugate kernel [25]. Hence, computing the meta-gradient is simply back-propagating through the kernel and a fixed feature extractor. To alleviate overfitting, we propose to maintain a diverse pool of models instead of periodically training and resetting a single model as in prior work [7, 13, 18]. Intuitively, our algorithm targets the following question: what is the best data to train the linear classifier given the current feature extractor? Due to the diverse feature extractors we use, the distilled data generalize well to a wide range of model distributions.

**Summary of Contributions:**

- We propose an effective method for dataset distillation. Our method, named neural Feature Regression with Pooling (FRePo), achieves state-of-the-art results on various benchmark datasets with a 100x reduction in training time and a 10x reduction in GPU memory requirement. Our distilled data looks real (Figure 1) and transfers well to different architectures.

- We show that FRePo scales well to datasets with high-resolution images or complex label space. We achieve 7.5% top1 accuracy on ImageNet-1K [26] using only one image per class. The same classifier obtains only 1.1% accuracy from a random subset of real images. The previous methods struggle in this task due to large memory and compute requirements.

- We demonstrate that high-quality distilled data can significantly improve various downstream applications, such as continual learning and membership inference defense.

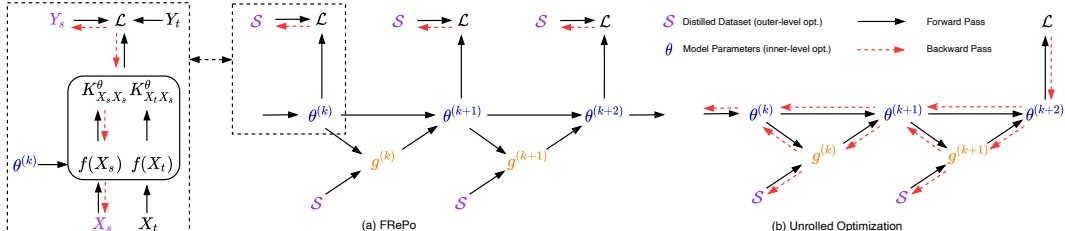

Figure 2: Comparison of FRePo and Unrolled Optimization. $S$, $X_s$, $Y_s$ are the distilled dataset, images and labels. $\mathcal{L}$ is the meta-training loss and $\theta^{(k)}$, $g^{(k)}$ are the model parameter and gradient at step $k$. $f(X)$ is the feature for input $X$ and $K^\theta_{X_t X_s}$ is the Gram matrix of $X_t$ and $X_s$. FRePo is analogous to 1-step TBPTT as it computes the meta-gradient at each step while performing the online model update. However, instead of backpropagating through the inner optimization, FRePo computes the meta-gradient through a kernel and feature extractor.

## 2 Method

### 2.1 Dataset Distillation as Bi-level Optimization

Suppose we have a large labeled dataset $\mathcal{T} = \left\{ (\mathbf{x}_1, \mathbf{y}_1), \ldots, (\mathbf{x}_{|\mathcal{T}|}, \mathbf{y}_{|\mathcal{T}|}) \right\}$ with $|\mathcal{T}|$ image and label pairs. Dataset distillation aims to learn a small synthetic dataset $\mathcal{S} = \left\{ (\mathbf{x}_1, \mathbf{y}_1), \ldots, (\mathbf{x}_{|\mathcal{S}|}, \mathbf{y}_{|\mathcal{S}|}) \right\}$ that preserves most of the information in $\mathcal{T}$. We train several neural networks parameterized by $\theta$ on the dataset $\mathcal{S}$ and then compute the validation loss $\mathcal{L}(\mathcal{A}lg(\theta, \mathcal{S}), \mathcal{T})$ on the real dataset $\mathcal{T}$, where $\mathcal{A}lg(\theta, \mathcal{S})$ is the neural network parameters optimized by a learning algorithm $\mathcal{A}lg$ with the model initialization $\theta$ and distilled dataset $\mathcal{S}$ as its inputs. The validation loss $\mathcal{L}(\mathcal{A}lg(\theta, \mathcal{S}), \mathcal{T})$ is a noisy objective with the stochasticity coming from random model initialization and inner learning algorithm. Thus, we are interested in minimizing the expected value of this loss, which we denote it as $F(\mathcal{S})$. We formulate the dataset distillation as the following bi-level optimization problem.

$$\overbrace{\mathcal{S}^* := \underset{\mathcal{S}}{\arg\min}\, F(\mathcal{S})}^{\text{outer-level}}, \quad \text{where } F(\mathcal{S}) = \mathbb{E}_{\theta \sim P_\theta}\left[ \mathcal{L}\Big( \overbrace{\mathcal{A}lg(\theta, \mathcal{S})}^{\text{inner-level}}, \mathcal{T} \Big) \right]. \tag{1}$$

In this bi-level setup, the outer loop optimizes the distilled data to minimize $F(\mathcal{S})$, while the inner loop trains a neural network using the learning algorithm, $\mathcal{A}lg$, to minimize the training loss on the distilled data $\mathcal{S}$. From the meta-learning perspective, the task is defined by the model initialization $\theta$, and we want to learn a meta-parameter $\mathcal{S}$ that generalizes well to different models sampled from the model distributions $P_\theta$. During learning, we optimize the meta-parameter $\mathcal{S}$ by minimizing the meta-training loss $F(\mathcal{S})$. In contrast, at meta-test time, we train a new model from scratch on $\mathcal{S}$ and evaluate the trained model on a held-out real dataset. This meta-test performance reflects the quality of the distilled data.

### 2.2 Dataset Distillation using Neural Feature Regression with Pooling (FRePo)

The outer-level problem can be solved using gradient-based methods of the form $\mathcal{S} \leftarrow \mathcal{S} - \alpha\,\nabla_{\mathcal{S}} F(\mathcal{S})$, where $\alpha$ is the learning rate for the distilled data and $\nabla_{\mathcal{S}} F(\mathcal{S})$ is the meta-gradient [27]. For a particular model $\theta$, the meta-gradient can be expressed as $\nabla_{\mathcal{S}} \mathcal{L}(\mathcal{A}lg(\theta, \mathcal{S}), \mathcal{T})$. Computing this meta-gradient requires differentiating through inner optimization. If $\mathcal{A}lg$ is an iterative algorithm like gradient descent, then backpropagating through the unrolled computation graph [14] can be a solution. However, this type of unrolled optimization introduces significant computation and memory overhead, as the whole training trajectory needs to be stored in memory (Figure 2(b)).

Traditionally, these issues are alleviated with truncated backpropagation through time (TBPTT) [28–30]. Instead of backpropagating through an entire unrolled sequence, TBPTT performs backpropagation for each subsequence separately. It is efficient because its time and memory complexity scale linearly with respect to the truncation steps. However, truncation may yield highly biased gradients that severely impact training. To mitigate this truncation bias [14], we consider *training only the top layer of a network to convergence*. The key insight is that the data helpful for training the output layer can also help train the whole network. Thus, we decompose the neural network into a feature

---

**Algorithm 1** Dataset Distillation using Neural Feature Regression with Pooling (FRePo)

---
**Require:** $\mathcal{T}$: a labeled dataset; $\alpha$: the learning rate for the distilled data
**Initialization:** Initialize a labeled distilled dataset $\mathcal{S} = (X_s, Y_s)$.
**Initialization:** Initialize a model pool $\mathcal{M}$ with $m$ models $\{\theta_i\}_{i=1}^{m}$ randomly initialized from $P_\theta$.
1: **while** not converged **do**
2:     $\triangleright$ Sample a model uniformly from the model pool: $\theta_i \sim \mathcal{M}$.
3:     $\triangleright$ Sample a target batch uniformly from the labeled dataset: $(X_t, Y_t) \sim \mathcal{T}$.
4:     $\triangleright$ Compute the meta-training loss $\mathcal{L}$ using Eq. 2
5:     $\triangleright$ Update the distilled data $\mathcal{S}$: $X_s \leftarrow X_s - \alpha \nabla_{X_s} \mathcal{L}$, and $Y_s \leftarrow Y_s - \alpha \nabla_{Y_s} \mathcal{L}$
6:     $\triangleright$ Train the model $\theta_i$ on the current distilled data $\mathcal{S}$ for one step.
7:     $\triangleright$ Reinitialize the model $\theta_i \sim P_\theta$ if $\theta_i$ has been updated more than $K$ steps.
8: **end while**
**Output:** Learned distilled dataset $\mathcal{S} = (X_s, Y_s)$

---

extractor and a linear classifier. We fix the feature extractor at each meta-gradient computation and train the linear classifier to convergence before updating $\mathcal{S}$. After that, we adjust the feature extractor by training the whole network on the updated distilled data. We note that similar two-phase procedure has been studied in the context of representation learning [31].

**Meta-Gradient Computation:** If we consider the mean square error loss, then the optimal weights for the linear classifier have a closed-form solution. Moreover, since the feature dimension is typically larger than the number of distilled data, we can use kernel ridge regression (KRR) with a conjugate kernel [25] rather than solving the weights explicitly [12]. The resulting meta-training loss (Eq. 2) is similar to that used in KIP [22, 23], but we use a more flexible kernel rather than NTK.

$$\mathcal{L}\left(Alg\left(\theta, \mathcal{S}\right), \mathcal{T}\right) = \frac{1}{2}||Y_t - K^{\theta}_{X_t X_s}(K^{\theta}_{X_s X_s} + \lambda I)^{-1} Y_s||_2^2, \tag{2}$$

where $(X_t, Y_t)$ and $(X_s, Y_s)$ are the inputs and labels of the real data $\mathcal{T}$ and distilled data $\mathcal{S}$ respectively. The Gram matrix between real inputs and distilled inputs is denoted as $K^{\theta}_{X_t X_s} \in \mathbb{R}^{|\mathcal{T}| \times |\mathcal{S}|}$, while the Gram matrix between distilled inputs is denoted as $K^{\theta}_{X_s X_s} \in \mathbb{R}^{|\mathcal{S}| \times |\mathcal{S}|}$. $\lambda$ controls the regularization strength for KRR. Let us denote the neural network feature for a given input $X$ and model parameter $\theta$ as $f(X, \theta) \in \mathbb{R}^{N \times d}$, where $N$ is the number of input and $d$ is the feature dimension [1]. The conjugate kernel is defined by the inner product of the neural network features. Thus, the two Gram matrices are computed as follows:

$$K^{\theta}_{X_t X_s} = f(X_t, \theta)f(X_s, \theta)^{\top}, \quad K^{\theta}_{X_s X_s} = f(X_s, \theta)f(X_s, \theta)^{\top}, \tag{3}$$

Now, computing the meta-gradient $\nabla_{\mathcal{S}} \mathcal{L}\left(Alg\left(\theta, \mathcal{S}\right), \mathcal{T}\right)$ is just back-propagating through the conjugate kernel and a fixed feature extractor, which is very efficient and takes even fewer operations than computing the gradient for the network's weights. Moreover, we decouple the meta-gradient computation from the model online update. Hence, we can train the online model using any optimizer, and the distilled data will be agnostic to the specific learning algorithm choice. Our proposed method is similar to 1-step TBPTT in that we compute the meta-gradient at each step while performing the online model update. Unlike the conventional 1-step TBPTT, we compute the meta-gradient using a KRR output layer to mitigate truncation bias, illustrated in Figure 2(a).

**Model Pool:** As discussed in Section 1, there are various types of overfitting in dataset distillation. Several techniques have been proposed to alleviate such problem, such as random initialization [4], periodic reset [13, 5, 7], and dynamic bi-level optimization [19]. These techniques share the same underlying principle: the model diversity matters. Thus, we propose to *maintain a "model pool" filled with diverse set of parameters* obtained from different number of training steps and different random initializations. Unlike the previous methods that periodically training and resetting a single model, FRePo randomly sample a model from the pool at each meta-gradient computation and update it using the current distilled data. However, if a model has been updated more than $K$ steps, we reinitialize it with a new random seed. From the meta-learning perspective, we maintain a diverse

---
[1] In practice, we use all the synthetic data and sample a minibatch from the real dataset to compute the meta-gradient (Algorithm 1).

set of meta-tasks to sample from and avoid sampling very similar tasks at each consecutive gradient computation to avoid overfitting to a particular setup.

**Pool Diversity:** We can increase the regularization strength by increasing the diversity of the model pool by setting a larger $K$, using data augmentation when training the model on the distilled data, or using models with different architectures. To keep our method simple, we use the same architecture for all models in the pool and do not use any data augmentation when training the model on the distilled data. Thus, our model pool only contains models with different initialization, at different optimization stages, and trained at different time-step of the distilled data.

## 3 Related Work

**Unrolling in Bi-Level Optimization:** One way to compute the meta-gradient is to differentiate through the unrolled inner optimization [4, 11–13]. However, this approach inherits several difficulties of the unrolled optimization, such as: 1) large computation and memory cost [14]; 2) truncation bias with short unrolls [17]; 3) exploding or vanishing gradients with long unrolls [15]; 4) chaotic and poorly conditioned loss landscapes with long unrolls [16]. In contrast, our method considers approximating the inner optimization with kernel ridge regression instead of unrolled optimization.

**Surrogate Objective:** To avoid unrolled optimization, several works turn to surrogate objectives. DC [5], DSA [7], and DCC [18] formulate the dataset distillation as a gradient matching problem between the gradients of neural network weights computed on the real and distilled data. In contrast, DM [8] and CAFE [19] consider the feature distribution alignment between the real and distilled data. Moreover, MTT [20] shows that knowledge from many expert training trajectories can be distilled to a dataset by using a training trajectory matching objective. Nevertheless, surrogate objectives may introduce new biases and thus, may not accurately reflect the true objective. For example, gradient matching approaches [5, 7, 18] only focus on short-range behavior and may easily overfit to a biased set of samples that produce dominant gradients [19, 20].

**Closed-form Approximation:** An alternative way to circumvent unrolled optimization is to find a closed-form approximation to the inner optimization. Based on the correspondence between infinitely-wide neural networks and kernel methods, KIP [22, 23] approximates the inner optimization with NTK [21]. In this case, the meta-gradient can be computed by back-propagating through the NTK. However, computing NTK for modern neural networks is extremely expensive. Thus, using NTK for dataset distillation requires thousands of GPU hours and sophisticated implementation of the distributed kernel computation framework [23]. Similar to ours, Bohdal et al. [12] also decomposes the neural network as a feature extractor and a linear classifier. However, they only learn the label and explicitly solve for the optimal classifier weights rather than perform KRR.

## 4 Dataset Distillation

### 4.1 Implementation Details

We compare our method to four state-of-the-art dataset distillation methods [7, 8, 20, 23] on various benchmark datasets [26, 32–37]. We train the distilled data using Algorithm 1 with the same set of hyperparameters for all experiments except stated otherwise. Unlike prior work [7, 8, 20], we do not apply data augmentation during training. However, we apply the same data augmentation [7, 20] during evaluation for a fair comparison. We preprocess the data in a similar way as in previous works [20, 23] but use a wider architecture than previous works [7, 8, 20] because the KRR component does not behave well when the feature dimension is low, resulting in a significant performance drop for our method. Results on the original architecture are included in Appendix **??**. We evaluate each distilled data using five random neural networks and report the mean and standard deviation. For the baseline method, we report the best of the reported value in the original paper and our reproducing results.

For the sake of brevity, we provide implementation details about data preprocessing, distilled data initialization, and hyperparameters in Appendix **??** and various ablation studies regarding the model pool, batch size, distilled data initialization, label learning, and model architectures in Appendix **??**. More distilled image visualizations can be found in Appendix **??**. Our code is available at https://github.com/yongchao97/FRePo.

Table 1: Test accuracies of models trained on the distilled data from scratch. $^\dagger$ denotes performance better than the original reported performance. KRR preformance is shown in bracket. FRePo performs extremely well for one image per class setting on CIFAR100, Tiny ImageNet and CUB-200.

| | Img/Cls | DSA [7] | DM [8] | KIP [23] | MTT [20] | FRePo |
|---|---|---|---|---|---|---|
| MNIST | 1 | $88.7 \pm 0.6$ | $89.9 \pm 0.8^\dagger$ | $90.1 \pm 0.1$ | $91.4 \pm 0.9^\dagger$ | $\mathbf{93.0 \pm 0.4}$ $(92.6 \pm 0.4)$ |
| | 10 | $97.9 \pm 0.1^\dagger$ | $97.6 \pm 0.1^\dagger$ | $97.5 \pm 0.0$ | $97.3 \pm 0.1^\dagger$ | $\mathbf{98.6 \pm 0.1}$ $(98.6 \pm 0.1)$ |
| | 50 | $\mathbf{99.2 \pm 0.1}$ | $98.6 \pm 0.1$ | $98.3 \pm 0.1$ | $98.5 \pm 0.1^\dagger$ | $\mathbf{99.2 \pm 0.0}$ $(99.2 \pm 0.1)$ |
| F-MNIST | 1 | $70.6 \pm 0.6$ | $71.5 \pm 0.5^\dagger$ | $73.5 \pm 0.5$ | $75.1 \pm 0.9^\dagger$ | $\mathbf{75.6 \pm 0.3}$ $(77.1 \pm 0.2)$ |
| | 10 | $84.8 \pm 0.3^\dagger$ | $83.6 \pm 0.2^\dagger$ | $86.8 \pm 0.1$ | $\mathbf{87.2 \pm 0.3}^\dagger$ | $86.2 \pm 0.2$ $(86.8 \pm 0.1)$ |
| | 50 | $88.8 \pm 0.2^\dagger$ | $88.2 \pm 0.1^\dagger$ | $88.0 \pm 0.1$ | $88.3 \pm 0.1^\dagger$ | $\mathbf{89.6 \pm 0.1}$ $(89.9 \pm 0.1)$ |
| CIFAR10 | 1 | $36.7 \pm 0.8^\dagger$ | $31.0 \pm 0.6^\dagger$ | $\mathbf{49.9 \pm 0.2}$ | $46.3 \pm 0.8$ | $46.8 \pm 0.7$ $(47.9 \pm 0.6)$ |
| | 10 | $53.2 \pm 0.8^\dagger$ | $49.2 \pm 0.8^\dagger$ | $62.7 \pm 0.3$ | $65.3 \pm 0.7$ | $\mathbf{65.5 \pm 0.4}$ $(68.0 \pm 0.2)$ |
| | 50 | $66.8 \pm 0.4^\dagger$ | $63.7 \pm 0.5^\dagger$ | $68.6 \pm 0.2$ | $71.6 \pm 0.2$ | $\mathbf{71.7 \pm 0.2}$ $(74.4 \pm 0.1)$ |
| CIFAR100 | 1 | $16.8 \pm 0.2^\dagger$ | $12.2 \pm 0.4^\dagger$ | $15.7 \pm 0.2$ | $24.3 \pm 0.3$ | $\mathbf{28.7 \pm 0.1}$ $(32.3 \pm 0.1)$ |
| | 10 | $32.3 \pm 0.3$ | $29.7 \pm 0.3$ | $28.3 \pm 0.1$ | $40.1 \pm 0.4$ | $\mathbf{42.5 \pm 0.2}$ $(44.9 \pm 0.2)$ |
| | 50 | $42.8 \pm 0.4$ | $43.6 \pm 0.4$ | – | $\mathbf{47.7 \pm 0.2}$ | $44.3 \pm 0.2$ $(43.0 \pm 0.3)$ |
| T-ImageNet | 1 | $6.6 \pm 0.2^\dagger$ | $3.9 \pm 0.2$ | – | $8.8 \pm 0.3$ | $\mathbf{15.4 \pm 0.3}$ $(19.1 \pm 0.3)$ |
| | 10 | – | $12.9 \pm 0.4$ | – | $23.2 \pm 0.2$ | $\mathbf{25.4 \pm 0.2}$ $(26.5 \pm 0.1)$ |
| CUB-200 | 1 | $1.3 \pm 0.1^\dagger$ | $1.6 \pm 0.1^\dagger$ | – | $2.2 \pm 0.1^\dagger$ | $\mathbf{12.4 \pm 0.2}$ $(13.7 \pm 0.2)$ |
| | 10 | $4.5 \pm 0.3^\dagger$ | $4.4 \pm 0.2^\dagger$ | – | – | $\mathbf{16.8 \pm 0.1}$ $(16.1 \pm 0.3)$ |

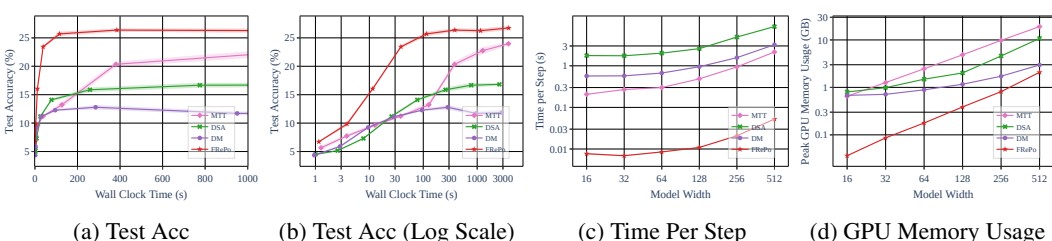

| (a) Test Acc | (b) Test Acc (Log Scale) | (c) Time Per Step | (d) GPU Memory Usage |
|---|---|---|---|

Figure 3: (a,b) Training efficiency comparison when learning 1 Img/Cls on CIFAR100. (c,d) Time per iteration and peak memory usage as we increase the model size. FRePo is significantly more efficient than the previous methods, almost two orders of magnitude faster than the second-best method (i.e., MTT), with only 1/10 of the GPU memory requirement.

## 4.2 Standard Benchmarks

**Distillation Performance:** We first evaluate our method on six standard benchmark datasets. We learn 1, 10, and 50 images per class for datasets with only ten classes, while we learn 1 and 10 images per class for CIFAR100 [34] with 100 classes, Tiny ImageNet [35] with 200 classes, and CUB-200 [37] with 200 fine-grained classes. As shown in Table in 1, we achieve the state-of-the-art performance in most settings despite the hyperparameter may be suboptimal. Our method performs exceptionally well on datasets with a complex label space when learning few images per class. For example, we improve the CIFAR100, Tiny ImageNet, and CUB-200 in one image per class setting from 24.3%, 8.8%, and 2.2% to 28.7%, 15.4%, and 12.4%, respectively. Figure 4 shows that our distilled images look real and natural though we do not directly optimize for this objective. We observe a strong correlation between the test accuracy and image quality: the better the image quality, the higher the test accuracy. Our results suggest that a highly condensed dataset does not need to be very different from the real dataset as it may just reflect the most common pattern in a dataset. We also report the KRR predictor's test accuracy using the feature extractor trained on the distilled data. When the dataset is as simple as MNIST [32], the KRR predictor achieves similar performance as the neural network predictor. In contrast, for more complex datasets, the KRR predictor consistently outperforms the neural network predictor, with the most significant gap being 3.7% for Tiny ImageNet in the one image per class setting.

Table 2: Cross-architecture transfer performance on CIFAR10 with 10 Img/Cls. Despite being trained for a specific architecture, our distilled data transfer well to various architectures unseen during training. Conv is the default evaluation model used for each method. NN, DN, IN, and BN stand for no normalization, default normalization, Instance Normalization, Batch Normalization respectively.

| | Train Arch | Evaluation Architecture | | | | | |
| --- | --- | --- | --- | --- | --- | --- | --- |
| | | Conv | Conv-NN | ResNet-DN | ResNet-BN | VGG-BN | AlexNet |
| DSA [7] | Conv-IN | $53.2 \pm 0.8$ | $36.4 \pm 1.5$ | $42.1 \pm 0.7$ | $34.1 \pm 1.4$ | $46.3 \pm 1.3$ | $34.0 \pm 2.3$ |
| DM [8] | Conv-IN | $49.2 \pm 0.8$ | $35.2 \pm 0.5$ | $36.8 \pm 1.2$ | $35.5 \pm 1.3$ | $41.2 \pm 1.8$ | $34.9 \pm 1.1$ |
| MTT [20] | Conv-IN | $64.4 \pm 0.9$ | $41.6 \pm 1.3$ | $49.2 \pm 1.1$ | $42.9 \pm 1.5$ | $46.6 \pm 2.0$ | $34.2 \pm 2.6$ |
| KIP [23] | Conv-NTK | $62.7 \pm 0.3$ | $58.2 \pm 0.4$ | $49.0 \pm 1.2$ | $45.8 \pm 1.4$ | $30.1 \pm 1.5$ | $57.2 \pm 0.4$ |
| FRePo | Conv-BN | $\mathbf{65.5 \pm 0.4}$ | $\mathbf{65.5 \pm 0.4}$ | $\mathbf{58.1 \pm 0.6}$ | $\mathbf{57.7 \pm 0.7}$ | $\mathbf{59.4 \pm 0.7}$ | $\mathbf{61.9 \pm 0.7}$ |

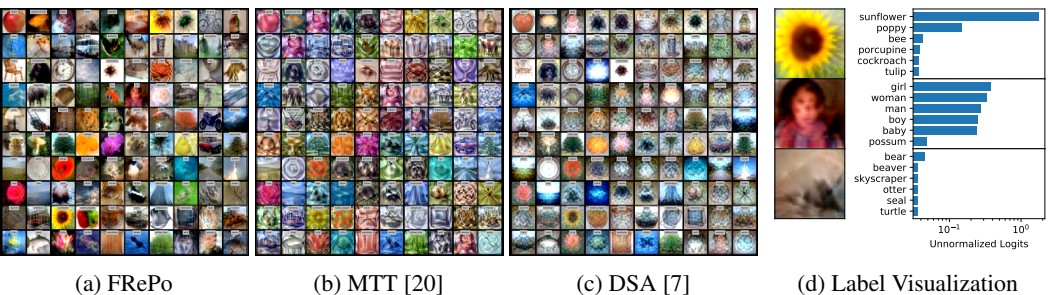

(a) FRePo      (b) MTT [20]      (c) DSA [7]      (d) Label Visualization

Figure 4: (a,b,c) Distilled 1 img/cls from CIFAR100 using FRePo, MTT, and DSA. High quality images also produce high test accuracy. (d) Three categories of learned labels. (Top) High confidence, large margin; (Middle) High confidence, small margin; (Bottom) Low confidence, small margin.

**Label Learning:** A similar trend can also be observed for label learning. When the dataset is simple and has only a few classes, label learning may not be necessary. However, it becomes crucial for complex datasets with many labels, such as CIFAR100 and Tiny-ImageNet (See more details in Appendix **??**). Similar to the teacher label in the knowledge distillation [1], we observe that the distilled label also encodes the class similarity. We identify three typical cases in Figure 4d. The first group consists of highly confident labels with a much higher value for one class than other classes (large margin), such as sunflower, bicycle, and chair. In contrast, the distilled labels in the second group are confident but may get confused with some closely-related classes (small margin). For instance, the learned label for "girl" has almost equally high values for the girl, woman, man, boy, and baby, suggesting that these classes are very similar and may be difficult for the model to distinguish them apart. The last group contains distilled labels with low values for all classes, such as bear, beaver, and squirrel. It is often hard for humans to recognize the distilled images in such a group, suggesting that they may be the challenging classes in a dataset.

**Training Cost Analysis:** Figure 3a, 3b shows that our method is significantly more time-efficient than the previous methods. When learning one image per class on CIFAR100, FRePo reaches a similar test accuracy (23.4%) to the second-best method (24.0%) in 38 seconds, compared to 3805 seconds for MTT, which is roughly two orders of magnitude faster. Moreover, FRePo achieves 92% of its final test accuracy (26.4% out of 28.7%) in only 385 seconds. As shown in Figure 3c, our algorithm takes much less time to perform one gradient step on the distilled data. Thus, we can perform more gradient steps in a fixed time. Furthermore, Figure 3d suggests that our algorithm has much less GPU memory requirement. Therefore, we can potentially use a much larger and more complex model to take advantage of the advancement in neural network architecture.

**Cross-Architecture Generalization:** One desired property of our distilled data is that it generalizes well to architecture it has not seen during the training. Similar to previous works [5, 20], we evaluate the distilled data from CIFAR10 on a wide range of architectures which it has not seen during training, including AlexNet [38], VGG [39], and ResNet [40]. Table 2 shows that our method outperforms previous methods on all unseen architectures. Instance Normalization (IN) [41], as the vital ingredient in several methods (DSA, DM, MTT), seems to hurt the cross-architecture transfer. The performance

Table 3: Distillation performance on higher resolution (128x128) dataset (i.e. ImageNette, Image-Woof) and medium resolution (64x64) dataset with a complex label space (i.e. ImageNet-1K). FRePo scales to high-resolution images and learns the discriminate feature of complex datasets.

| Img/Cls | ImageNette (128x128) | | ImageWoof (128x128) | | ImageNet (64x64) | |
|---|---|---|---|---|---|---|
| | 1 | 10 | 1 | 10 | 1 | 2 |
| Random Subset | $23.5 \pm 4.8$ | $47.7 \pm 2.4$ | $14.2 \pm 0.9$ | $27.0 \pm 1.9$ | $1.1 \pm 0.1$ | $1.4 \pm 0.1$ |
| MTT [20] | $47.7 \pm 0.9$ | $63.0 \pm 1.3$ | $28.6 \pm 0.8$ | $35.8 \pm 1.8$ | $-$ | $-$ |
| FRePo | $\mathbf{48.1 \pm 0.7}$ | $\mathbf{66.5 \pm 0.8}$ | $\mathbf{29.7 \pm 0.6}$ | $\mathbf{42.2 \pm 0.9}$ | $\mathbf{7.5 \pm 0.3}$ | $\mathbf{9.7 \pm 0.2}$ |

degrades a lot when no normalization (NN) is applied (Conv-NN, AlexNet) or using a different normalization, like Batch Normalization (BN) [42]. It suggests that the distilled data generated by those methods encode the inductive bias of a particular training architecture. In contrast, our distilled data generalize well to various architectures, including those without normalization (Conv-NN, AlexNet). Note that Figure 1, 4 also indicate that our distilled data encode less architectural bias as the distilled images look natural and authentic. A simple idea to further alleviate the overfitting of a particular architecture is to include more architectures in the model pool. However, the training may not be stable as the meta-gradient computed by different architectures can be very different.

## 4.3 ImageNet

**High Resolution ImageNet Subset** To understand how well our method performs on high-resolution images, we evaluate it on ImageNette and ImageWoof datasets [36] with a resolution of 128x128. We learn 1 and 10 images per class on both datasets and report the performance in Table 3 and visualize some distilled images in Figure 1. As shown in Table 3, we outperform MTT on all settings and achieve much better performance when we distill ten images per class on a more difficult dataset ImageWoof. It suggests that our distilled data is better at capturing the discriminative features for each class. Figure 1 shows that our distilled images look real and capture the distinguishable feature of different classes. For the easy dataset (i.e., ImageNette), all images have clear different structures, while for ImageWoof, the texture of each dog seems to be crucial.

**Resized ImageNet-1K:** We also evaluate our method on a resized version of ILSVRC2012 [26] with a resolution of 64x64 to see how it performs on a complex label space. Surprisingly, we can achieve 7.5% and 9.7% Top1 accuracy using only 1k and 2k training examples, compared to 1.1% and 1.4% using an equally-sized real subset.

## 5 Application

### 5.1 Continual Learning

Continual learning (CL) [43] aims to address the catastrophic forgetting problem [43–45] when a model learns sequentially from a stream of tasks. A commonly used strategy to recall past knowledge is based on a replay buffer, which stores representative samples from previous tasks [46–49]. Since sample selection is an important component of constructing an effective buffer [48–51], we believe distilled data can be a key ingredient for a continual learning algorithm due to its highly condensed nature. Several works [6–8, 52] have successfully applied the dataset distillation to the continual learning scenario. Our work shows that we can achieve much better results by using a better dataset distillation technique.

We follow Zhao and Bilen [8] that sets up the baseline based on GDumb [49] which greedily stores class-balanced training examples in memory and train model from scratch on the latest memory only. In that case, the continual learning performance only depends on the quality of the replay buffer. We perform 5 and 10 step class-incremental learning [53] on CIFAR100 with an increasing buffer size of 20 images per class. Specifically, we distill 400 and 200 images at each step and put them into the replay buffer. We follow the same class split as Zhao and Bilen [8] and compare our method to random [49], herding [54, 55], DSA [7], and DM [8]. We use the default data preprocessing and default model for each method in this experiment as we find it gives the best performance for each method. We use the test accuracy on all observed classes as the performance measure [8, 48].

Table 4: AUC of five attackers on models trained on the real and distilled MNIST data. The model trained on the real data is vulnerable to MIAs, while the model trained on the distilled data is robust to MIAs. Training on distilled data allows privacy preservation while retaining model performance.

| | Test Acc (%) | Attack AUC | | | | |
| | | Threshold | LR | MLP | RF | KNN |
|---|---|---|---|---|---|---|
| Real | **99.2 ± 0.1** | 0.99 ± 0.01 | 0.99 ± 0.00 | 1.00 ± 0.00 | 1.00 ± 0.00 | 0.97 ± 0.00 |
| Subset | 96.8 ± 0.2 | 0.52 ± 0.00 | **0.50 ± 0.01** | **0.53 ± 0.01** | 0.55 ± 0.00 | 0.54 ± 0.00 |
| DSA | 98.5 ± 0.1 | **0.50 ± 0.00** | 0.51 ± 0.00 | 0.54 ± 0.00 | 0.54 ± 0.01 | 0.54 ± 0.01 |
| DM | 98.3 ± 0.0 | **0.50 ± 0.00** | 0.51 ± 0.01 | 0.54 ± 0.01 | 0.54 ± 0.01 | 0.53 ± 0.01 |
| FRePo | 98.5 ± 0.1 | 0.52 ± 0.00 | 0.51 ± 0.00 | **0.53 ± 0.01** | **0.52 ± 0.01** | **0.51 ± 0.01** |

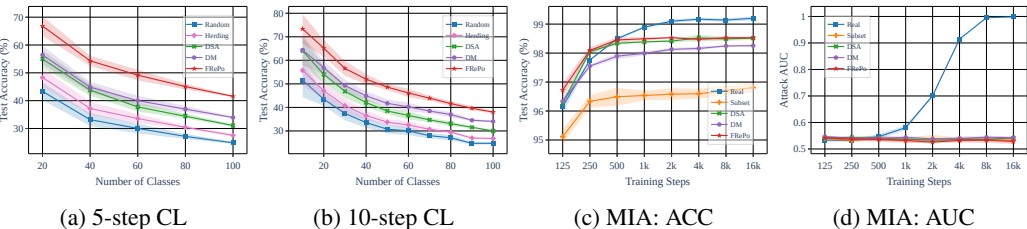

| (a) 5-step CL | (b) 10-step CL | (c) MIA: ACC | (d) MIA: AUC |

Figure 5: (a,b) Multi-class accuracies across all classes observed up to a certain time point. We perform significantly better than other methods in both 5 and 10 step class-incremental continual learning. (c,d) Test accuracy and attack AUC as we increase the number of training steps. AUC keeps increasing when training a model on the real data for more steps. In contrast, AUC keeps low when training on distilled data.

Figure 5 shows that our method performs significantly better than all previous methods. The final test accuracy for all classes for our method (FRePo) and the second-best method (DM) are 41.6%, 33.9% in 5-step learning, and 38.0%, 34.0% in 10-step learning. However, we notice that for FRePo, distilling 2000 images in a continual learning setup achieves a similar test accuracy (41.6%) as distilling only 1000 images from the whole dataset (41.3% from Table 1). In addition, performance drops as we perform more steps. It suggests that FRePo considers all available classes to derive the most condensed dataset. Splitting the data into multiple groups and performing independent distillation may generate redundant information or fail to capture the distinguishable features.

## 5.2 Membership Inference Defense

Membership inference attacks (MIA) aim to infer whether a given data point has been used to train the model or not [56–58]. Ideally, we want a model to learn from the data but not memorize it to preserve privacy. However, deep neural networks are well-known for their ability to memorize all the training examples, even on large and randomly labeled datasets [59]. Several methods have been proposed to defend against such attacks by either modifying the training procedure [60] or changing the inference workflow [61]. This section shows that the distilled data contain little information regarding sample presence in the original dataset. Thus, instead of training on the original datasets, training on distilled data allows privacy preservation while retaining model performance.

We consider three distilled data generated by DSA [7], DM [8] and FRePo. We perform five popular "black box" MIA provided by Tensorflow Privacy [62] on models trained on the real data or the data distilled from it. The attack methods include a threshold attack and four model-based attacks using logistic regression (LR), multi-layer perceptron (MLP), random forest (RF) and K-nearest neighbor (KNN). The inputs to those attack methods are ground-truth labels, model predictions, and losses. To measure the privacy vulnerability of the trained model, we compute the area under the ROC curve (AUC) of an attack classifier. Following prior work, [56, 63], we keep a balanced set of training examples (member) and test examples (non-member) with 10K each to maximize the uncertainty of MIA. Thus, the random guessing strategy results in a 50% MIA accuracy. We conduct experiments on MNIST and FashionMNIST with a distillation size of 500. For space reasons, we provide more implementation details and results in appendix.

As shown in Table 4, all models trained on the distilled data preserve privacy as their attack AUCs are closed to random guessing. However, we observe a small drop in test accuracy compared to the model trained on the full dataset, which is expected as we only distill 500 examples instead of 10,000 examples. Compared to the model trained on an equally sized subset of the original data, the model trained on distilled data results in much better test performance. Figure 5c, 5d demonstrate the trade-off between test accuracy and attack effectiveness as measured by ROC AUC. It shows that early stopping can be an effective technique to preserve privacy. However, we will still be under high MIA risk if we perform early stopping by monitoring the validation loss. In contrast, training a model on the distilled data does not have this problem as the attack AUCs keep at a very low level regardless of training steps.

## 6 Conclusion

We propose neural Feature Regression with Pooling (FRePo) to overcome two challenges in dataset distillation: meta-gradient computation and various types of overfitting in dataset distillation. We obtain state-of-the-art performance on various datasets with a 100x reduction in training time and a 10x reduction in GPU memory requirement. The distilled data generated by FRePo looks real and natural and generalizes well to a wide range of architectures. Furthermore, we demonstrate two applications that take advantage of the high-quality distilled data, namely, continual learning and membership inference defense.

**Broader Impact** "Synthetic data", in the broader sense of artificial data created by generative models, can help researchers understand how an otherwise opaque learning machine "sees" the world. There have been concerns regarding the risk of fake data. This paper explores a new research direction in generating synthetic data only for downstream classification tasks. We believe this work can provide additional interpretability and potentially address the common concerns in machine learning regarding training data privacy.

## Acknowledgments and Disclosure of Funding

We would like to thank Harris Chan, Andrew Jung, Michael Zhang, Philip Fradkin, Denny Wu, Chong Shao, Leo Lee, Alice Gao, Keiran Paster, and Lazar Atanackovic for their valuable feedback. Jimmy Ba was supported by NSERC Grant [2020-06904], CIFAR AI Chairs program, Google Research Scholar Program and Amazon Research Award. This project was supported by LG Electronics Canada. Resources used in preparing this research were provided, in part, by the Province of Ontario, the Government of Canada through CIFAR, and companies sponsoring the Vector Institute for Artificial Intelligence.

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
