# A Experimental Details

## A.1 Implementation Details

**Datasets:** We evaluate our methods on the following datasets: i) **MNIST** [32]: A standard image dataset consists of 10 classes of 28x28 grey-scale images of handwritten digits, including 60,000 training examples and 10,000 test examples. ii) **FashionMNIST** [33]: A direct drop-in replacement for MNIST [32] consisting of 10 classes of 28x28 grey-scale images of clothing, including 60,000 training examples and 10,000 test examples. We denote FashionMNIST as F-MNIST for simplicity. iii) **CIFAR** [34]: A standard image dataset with two tasks: one coarse-grained over 10 classes (CIFAR10) and one fine-grained over 100 classes (CIFAR100). Both CIFAR10 and CIFAR100 have 50,000 training examples and 10,000 test examples with a resolution of 32x32. iv) **Tiny ImageNet** [35]: A higher resolution (64x64) dataset with 200 classes, including 100,000 training examples and 10,000 test examples. We denote Tiny ImageNet as T-ImageNet for simplicity. v) **ImageNette** and **ImageWoof** [36]: ImageNette (assorted objects) and ImageWoof (dog breeds) are two 10-class subsets from ILSVRC2012 [26] designed to be easy and hard to learn, respectively. ImageNette consists of 9,469 training examples and 3,925 testing examples, while ImageWoof contains 9025 training examples and 3929 testing examples. We resize all examples to a resolution of 128x128. vi) **ImageNet** [26]: A standard image benchmark dataset consists of 1000 classes from the Large Scale Visual Recognition Challenge 2012 [26], including 1,281,167 training examples and 50,000 testing examples. We resize all examples to a resolution of 64x64. vii) **CUB-200** [37]: A fine-grained classification task, consisting of 200 subcategories belonging to birds. It has 5,994 data points for training and 5,794 data points for testing. We resize all examples to a resolution of 32x32.

**Data Preprocessing:** We use the standard preprocessing for all datasets but add regularized ZCA transformation for RGB datasets as described in KIP [22, 23]. However, unlike KIP, we do not apply layer normalization to all examples. Moreover, for simplicity, we apply the same regularization strength $\lambda = 0.1$ to all datasets rather than tune each dataset. This regularization strength is tuned for CIFAR in KIP [23]. Besides, for ImageNette and ImageWoof, performing the full-size ZCA transformation is extremely expensive due to the high resolution, so we use a checkboard ZCA instead. It is the reason why our distilled images in Figure 1(c) have checkboard artifacts. As for the previous method, DSA [7], DM [8] only use standard preprocessing, while MTT [20] and KIP [23] consider both standard preprocessing and ZCA preprocessing, but they implement ZCA preprocessing differently. To account for the difference in data preprocessing, we reproduce each method using our data preprocessing and report the best of the reported value in the original paper and our reproducing results. As shown in Table 1, it turns out that our data preprocessing can improve DSA [7] and DM [8] a lot on RGB datasets but achieves a comparable performance for MTT [20]. When visualizing the distilled images, we directly apply the reverse transformation according to the corresponding data preprocessing without any other modifications.

**Models:** We use a simple convolutional neural network for all experiments based on the architecture used in previous works [7, 20, 23]. It consists of several blocks of 3x3 convolution, normalization, RELU, and 2×2 average pooling layer with stride 2. We use 3, 4, and 5 blocks for datasets with resolutions 32x32, 64x64, and 128x128, respectively. Unlike previous works, which use the same number of filters at every layer, we double the number of filters if the feature map size is halved, following the modern neural network designs that preserve the time complexity per layer[39, 40]. We observe this to be crucial for our method when we distill thousands of data because we need the feature dimension to be much larger than the distilled size to make the KRR work properly. Furthermore, we replace the Instance Normalization [41] with Batch Normalization [42] during training and do not use any normalization during evaluation. For DSA, DM, and MTT, the default normalization for both training and evaluation is the Instance Normalization [41]. In contrast, KIP [23] uses analytical NTK during training and uses a 1024-width network without normalization for evaluation. Moreover, KIP [23] adds an extra convolution layer at the first layer of the neural network. We initialize our model using Lecun Initialization, which is the default in Flax library [64]. In contrast, DSA, DM, and MTT use Kaiming Initialization, which is the default in PyTorch, while KIP [23] initializes the model using random Gaussian with standard deviations $\sqrt{2}$ and 0.1 for weights and biases, respectively. We denote the default model used in each method as Conv for simplicity. We reproduce each method using our model and report the best of the reported value in the original paper and our reproducing results. We find our model achieves comparable or worse performance than each method's default methods, so we use their default model in most settings.

---

**Algorithm 2** Dataset Distillation using Neural Feature Regression with Pooling (FRePo)

---

**Require:** $\mathcal{T}$: a labeled dataset; $\alpha$: the learning rate for the distilled data
**Initialization:** Initialize a labeled distilled dataset $\mathcal{S} = (X_s, Y_s)$.
**Initialization:** Initialize a model pool $\mathcal{M}$ with $m$ models $\{\theta_i\}_{i=1}^{m}$ randomly initialized from $P_\theta$.

1: **while** not converged **do**
2:     ▷ Sample a model uniformly from the model pool: $\theta_i \sim \mathcal{M}$.
3:     ▷ v1: Train the model $\theta_i$ on the current distilled data $\mathcal{S}$ for one step.
4:     ▷ Sample a target batch uniformly from the labeled dataset: $(X_t, Y_t) \sim \mathcal{T}$.
5:     ▷ Compute the meta-training loss $\mathcal{L}$ using Eq. 2
6:     ▷ Update the distilled data $\mathcal{S}$: $X_s \leftarrow X_s - \alpha \nabla_{X_s} \mathcal{L}$, and $Y_s \leftarrow Y_s - \alpha \nabla_{Y_s} \mathcal{L}$
7:     ▷ v2: Train the model $\theta_i$ on the current distilled data $\mathcal{S}$ for one step.
8:     ▷ Reinitialize the model $\theta_i \sim P_\theta$ if $\theta_i$ has been updated more than $K$ steps.
9: **end while**
**Output:** Learned distilled dataset $\mathcal{S} = (X_s, Y_s)$

---

Besides, we notice that the distilled data can encode the architecture's inductive bias, so we provide an ablation study on how the model architecture affects the distilled data in Section C.6.

**Initialization:** Staying consistent with previous works, [7, 20, 23], we initialize the distilled image with randomly sampled real images. We also investigate initializing the distilled image with random Gaussian noise and observe that initializing does not affect the final performance too much but affects the convergence speed. We find the real initialization gives a decent convergence speed, so we choose the real initialization for simplicity rather than fine-tune the scale of random Gaussian initialization. As for labels, we initialize them using a scaled mean-centered one-hot vector for the corresponding image, where the scaling factor (i.e., $1/(\sqrt{C/10})$) depends on the number of classes $C$. We find that this label initialization scheme speeds up the convergence but has little impact on the final performance. We provide an ablation study regarding the initialization scheme in Section C.3.

**Label Learning:** Whether to learn the label is a Boolean hyperparameter in our experiments. When true, we optimize it using Algorithm 1. When false, we stop the gradient so that they remain fixed at its initialization. We provide an ablation study on label learning in Section C.4.

**Training and Evaluation:** The proposed algorithm has two versions depending on the order of meta-gradient computation and online model update. We present the two versions in Algorithm 2 and highlight the difference in red. We denote computing the meta-gradient after the online model update as v1 and denote computing the meta-gradient before the online model update as v2. The first implementation (v1) is more similar to the standard back-propagation through time as we first do the forward pass (online model update) and then the backward pass. However, instead of backpropagating the gradient through the inner optimization, we backpropagate through the kernel. In contrast, the insight behind the second implementation (v2) is that we want to find good data to train the last layer first, then we use it to train the whole network. Empirically, we do not observe any difference between these two implementations, and we choose v2 as the default for all experiments. Besides, we use the distilled data and its flipped version in meta-gradient computation to mitigate the mirroring effect for RGB datasets. For evaluation, we follow the standard protocol [7, 20]: training a randomly initialized neural network from scratch on distilled data and evaluating on a held-out test dataset. We apply the same data augmentation as in previous work [7, 20] during evaluation for a fair comparison.

**Hyperparameters:** We aim to keep our method as simple and efficient as possible. As a result, our method requires very few hyperparameter tuning efforts than all the previous methods. We use the same set of hyperparameters for all experiments, except stated otherwise. Specifically, we use LAMB optimizer [65] with a cosine learning rate schedule starting from 0.0003 for both images and labels. We use a batch size of 1024 and train the distilled data up to 2 million steps to see its long-run behavior. In practice, we observe that most of the convergence (more than 95% of its final test accuracy) is achieved after a few thousand steps with a slow, logarithmic increase with more iterations, as shown in Figure 3b. The model pool contains ten models trained up to $K = 100$ steps. Each model is trained using Adam optimizer [66] with a constant learning rate of 0.0003, and no weight decay is applied. We use the same kernel regularizer $\lambda$ as KIP [23]. Instead of being a fixed constant, the regularizer is adapted to the scale of $K_{X_s X_s}^\theta$, $\lambda = \lambda_0 Tr(K_{X_s X_s}^\theta)$, where $\lambda_0 = 10^{-6}$. We note that our hyperparameter choice may be sub-optimal, but it is a good starting point. We conduct some ablation study regarding the hyperparameters in Section C. Moreover, we provide a

hyperparameter tuning guideline for practitioners in Section D accompanied by a list of additional tricks that we find to improve the performance but do not include in the current algorithm.

**Summary:** We implement our method in JAX [67] and reproduce previous methods using their released code. To take into account the differences in data processing and architectures, we try our best to reproduce previous results by varying different data preprocessing and models. Experiments show that our data preprocessing can sometimes improve performance, but our model does not give better performance than previous methods. We report the best of the reported value in the original paper and our reproducing results. We evaluate each distilled data using five random neural networks and report the mean and standard deviation.

## A.2 Experimental Setups

**Figure 1:** Selected images from (a) 1 Img/Cls CIFAR100 (ZCA, learn label=True), (b) 1 Img/Cls Tiny ImageNet (ZCA, learn label=True), (c) 10 Img/Cls ImageNette (Top 2 rows) (Checkboard ZCA, learn label=True), 10 Img/Cls ImageWoof (Bottom 2 rows) (Checkboard ZCA, learn label=True).

**Figure 3a, 3b:** We measure the time per step by measuring the average wall clock time ten times on ten steps. We take the first measurement after 50 steps when the statistics become stable and report the mean and standard deviation of 10 runs. To generate Figure 3a, 3b, we use the default model and default hyperparameter for each method. We first record the time per step of each method and run another program to evaluate the checkpoints at different time steps to get the test accuracy. Thus, the wall clock time is computed by multiplying the time per step and the number of steps taken at each checkpoint. All models are trained on Nvidia Quadro RTX 6000 with 22.17GB memory, and 7.5 compute capability.

**Figure 3c, 3d, 14a, 14b:** Similar to how we generate Figure 3a, 3b, we measure the time per step by measuring the average wall clock time ten times on ten steps. We take the first measurement after 50 steps when the statistics become stable and report the mean and standard deviation of 10 runs. We use jax.profiler and torch.profiler for the JAX program and PyTorch program to measure the peak GPU memory usage. Different from Figure 3a, 3b, we use the same model (i.e., Conv used by DSA [7]) and same batch size=256 in this section. We take an optimistic estimation for the previous methods (i.e., DSA, MTT) by choosing a smaller inner loop or outer loop number to generate more data points before encountering out of memory errors. Specifically, we use outer_loop=1, inner_loop=1 for DSA and use syn_steps=15 for MTT. All models are trained on Nvidia Quadro RTX 6000 with 22.17GB memory, and 7.5 compute capability.

**Figure 4a**: Distilled image visualization when distilling 1 Img/Cls from CIFAR100 using FRePo.

**Figure 4b**: Distilled image visualization when distilling 1 Img/Cls from CIFAR100 using MTT [20]. To give the best image quality, we use the distilled images provided by the original paper. (Url:`https://georgecazenavette.github.io/mtt-distillation/tensors/index.html#tensors`)

**Figure 4c**: Distilled image visualization when distilling 1 Img/Cls from CIFAR100 using DSA [7]. To give the best image quality, we use the distilled images provided by the original paper. (Url: `https://drive.google.com/drive/folders/1Dp6V6RvhJQPsB-2uZCwdlHXf1iJ9Wb_g`). Besides, we adjust the contrasts to give the best visualization.

**Figure 4d**: Distilled label visualization when distilling 1 Img/Cls from CIFAR100 using FRePo. The distilled images are sunflower, girl, and bear, respectively. All labels are unnormalized logits.

**Figure 5a, 5b:** Multi-class accuracies across all classes observed up to a specific time point. For a fair comparison, we follow the same setup in DM [8], including the class split of five different runs. We use the same set of hyperparameters as other experiments and train the distilled data up to 500K steps. We report the mean and standard deviation of the five different runs. We observe that the primary source of variance comes from the class split.

**Figure 5c, 5d, 6a, 6b:** Test accuracies and attack AUCs as we increase the number of training steps. We use the same set of hyperparameters as other experiments except for data augmentation. We do not apply any data augmentation when we train models on the distilled data since any type of regularization can alleviate the MIA risks, making it hard to see the effects of distilled data.

**Table 1, 3, 7, 5:** We try to reproduce the previous methods based on their official codebase and vary the data preprocessing and model architecture. We report the best of the reported value in the original

Table 5: Distillation performance on higher resolution (128x128) dataset (i.e. ImageNette, Image-Woof) and medium resolution (64x64) dataset with a complex label space (i.e. ImageNet-1K). FRePo scales well to high-resolution images and learns the discriminate feature of complex datasets well.

| Img/Cls | ImageNette (128x128) | | ImageWoof (128x128) | | ImageNet (64x64) | |
|---|---|---|---|---|---|---|
| | 1 | 10 | 1 | 10 | 1 | 2 |
| Random Subset | $23.5 \pm 4.8$ | $47.7 \pm 2.4$ | $14.2 \pm 0.9$ | $27.0 \pm 1.9$ | $1.1 \pm 0.1$ | $1.4 \pm 0.1$ |
| MTT [20] | $47.7 \pm 0.9$ | $63.0 \pm 1.3$ | $28.6 \pm 0.8$ | $35.8 \pm 1.8$ | $-$ | $-$ |
| FRePo | $\mathbf{48.1 \pm 0.7}$ | $\mathbf{66.5 \pm 0.8}$ | $\mathbf{29.7 \pm 0.6}$ | $\mathbf{42.2 \pm 0.9}$ | $\mathbf{7.5 \pm 0.3}$ | $\mathbf{9.7 \pm 0.2}$ |
| Full Dataset | $87.9 \pm 1.0$ | | $74.4 \pm 1.6$ | | $19.8 \pm 0.6$ | |

paper and our reproducing results. As for our method, Table 1, 3 report the best value of learning label or not learning label with our default architecture and data preprocessing. We use the same set of hyperparameters for all experiments. The complete results can be found in Table 7. Besides, we also include the results when training the model on the full dataset using mean square error loss in Table 5.

**Table 2:** We generate the distilled data for each method using their default model and default hyperparameter. For KIP, since the training is too expensive, we use the checkpoint provided by the original author. We perform a sweep on the checkpoints in "gs://kip-datasets/kip/cifar10/" and find "ConvNet_ssize100_zca_nol_noaug_ckpt1000.npz" gives the best performance that matches the reported value in the original paper. We evaluate DSA, DM, and MTT in PyTorch and KIP and FRePo in JAX. We notice that our reproducing results for KIP are much better than the reproducing results reported in MTT [20]. It may be due to the differences in initialization or hyperparameter choices, such as learning rate.

**Table 4, 6:** Table 4 and Table 6 present the test accuracies and MIA results on MNIST and Fashion-MNIST, respectively. We random sample 10K data points to generate the distilled data and sample another 10K non-overlapped data as non-member data. We use the same set of hyperparameters as other experiments except for data augmentation. We do not apply any data augmentation when we train models on the distilled data since any regularization can alleviate the MIA risks, making it hard to see the effects of distilled data.

## B  Additional Results

**Resized ImageNet-1K:** We also evaluate our method on a resized version of ILSVRC2012 [26] with a resolution of 64x64 to see how it performs on a complex label space. As shown in Table 5, we can achieve 7.5% and 9.7% Top1 accuracy using only 1k and 2k training examples, compared to 1.1% and 1.4% using an equally-sized real subset or 19.8% on the full dataset using 1281167 training examples. We notice that Mean Square Error (MSE) loss may not be suitable for a complex dataset like ImageNet as the same model trained with Cross-Entropy loss achieve a Top1 accuracy of 32.2%. However, since we use MSE loss to evaluate the distilled data, we report the model trained using MSE loss for a fair comparison.

**Membership Inference Defense:** We only show MNIST results in Section 5.2 due to space reasons. We provide the same experimental results on FashionMNIST in Figure 6 and Table 6. We observe a similar trend on the two different datasets: training a model on the distilled data can preserve the privacy while achieving a good performance. We notice that there is a gap between training on the whole data set, which may be closed by distilling more data points and adding more noise when perform the distillation.

## C  Ablation Study

We conduct several ablation studies to understand the key components of the proposed method, including the role of kernel approximation (Section C.1), model pool (Section C.2), initialization (Section C.3), label learning (Section C.4), scalability (Section C.5), and model architecture C.6.

Table 6: AUC of attacker classifier trained on real data and distilled data on FashionMNSIT. We highlight the best attacker performance in bold for each model. The model trained on real data is vulnerable to membership inference attacks. In contrast, the model trained on distilled data is robust to membership inference attacks.

| | Test Acc (%) | Attack AUC | | | | |
|---|---|---|---|---|---|---|
| | | Threshold | LR | MLP | RF | KNN |
| Real | $89.7 \pm 0.2$ | $0.99 \pm 0.01$ | $0.99 \pm 0.00$ | $0.99 \pm 0.00$ | $0.99 \pm 0.00$ | $0.98 \pm 0.00$ |
| Subset | $81.1 \pm 0.7$ | $0.53 \pm 0.01$ | $0.51 \pm 0.01$ | $0.52 \pm 0.01$ | $0.52 \pm 0.01$ | $0.53 \pm 0.00$ |
| DSA | $87.0 \pm 0.1$ | $0.51 \pm 0.00$ | $0.51 \pm 0.01$ | $0.51 \pm 0.01$ | $0.52 \pm 0.01$ | $0.51 \pm 0.01$ |
| DM | $87.3 \pm 0.1$ | $0.52 \pm 0.00$ | $0.51 \pm 0.01$ | $0.50 \pm 0.01$ | $0.52 \pm 0.01$ | $0.51 \pm 0.01$ |
| FRePo | $87.6 \pm 0.2$ | $0.52 \pm 0.00$ | $0.53 \pm 0.01$ | $0.53 \pm 0.01$ | $0.53 \pm 0.01$ | $0.52 \pm 0.00$ |

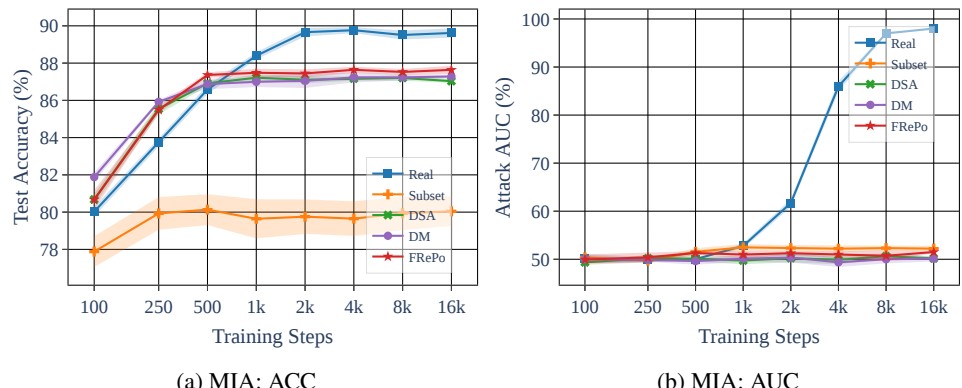

(a) MIA: ACC          (b) MIA: AUC

Figure 6: (a,b) Test accuracy and attack AUC on Fashion MNIST as we increase the number of training steps. AUC keeps increasing when training a model on the real data for more steps. In contrast, AUC keeps low when training on distilled data.

All experiments are conducted on CIFAR100 when learning 1 Img/Cls using Algorithm 1 with the default hyperparameters except stated otherwise.

## C.1 FRePo vs TBPTT

Figure 7 illustrates the computation graph of FRePoand 1-step TBPTT, which is different from Figure 2 where we compare FRePo with unrolled optimization. FRePo differs from 1-step TBPTT in meta-gradient computation. As shown in Figure 7, 1-step TBPTT computes the meta-gradient by backpropagating through the inner optimization, while FRePo uses backpropagating through a kernel and a feature extractor. Figure 8 compares the training loss, training accuracy, and test accuracy of FRePo and TBPTT when using the same training and evaluation protocol. Note that FRePo computes the training loss and accuracy using the KRR head during training, while TBPTT uses the neural network head. We use the same optimizer to perform the online model update and use the same optimizer to train the same neural network for the same amount of steps on the distilled data during evaluation.

Figure 8c shows that the distilled data generated by TBPTT keeps getting worse test accuracy as the training goes on. It suggests that the meta-gradient computed by TBPTT is highly biased and does not help learn a generalizable distilled dataset. As a result, the distilled data is overfitted to a k-step learning setup, where the k is the truncation step. This learning scenario is very similar to the dataset distillation setup in DD [4] and MTT [20], where a specific optimizer is learned to take advantage of the distilled data. Though using more truncation steps can alleviate this problem, eliminating the truncation bias needs an infinite unrolled optimization, making it intractable. In contrast, FRePo alleviates the truncation bias by training a subset of a neural network to convergence. Moreover, FRePo decouples the meta-gradient computation and online model update such that the distilled data will not overfit the inner-level optimization. Conversely, TBPTT needs to fine-tune the inner optimization to get the best performance, which we do not explore further here.

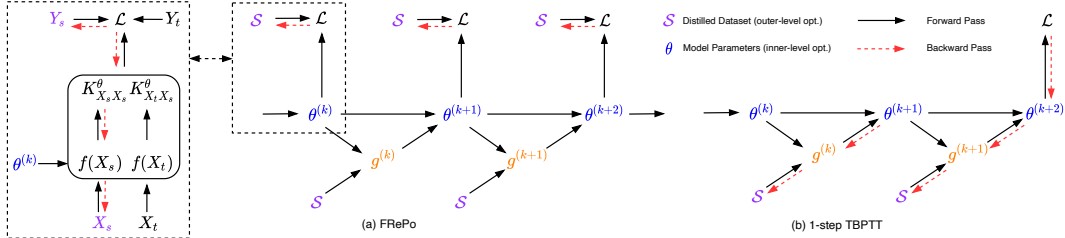

Figure 7: Comparison of FRePo and 1-step TBPTT. $S$, $X_s$, $Y_s$ are the distilled dataset, images and labels. $\mathcal{L}$ is the meta-training loss and $\theta^{(k)}$, $g^{(k)}$ are the model parameter and gradient at step $k$. $f(X)$ is the feature for input $X$ and $K^\theta_{X_t X_s}$ is the Gram matrix of $X_t$ and $X_s$. FRePo is analogous to 1-step TBPTT as it computes the meta-gradient at each step while performing the online model update. However, instead of backpropagating through the inner optimization, FRePo computes the meta-gradient through a kernel and feature extractor.

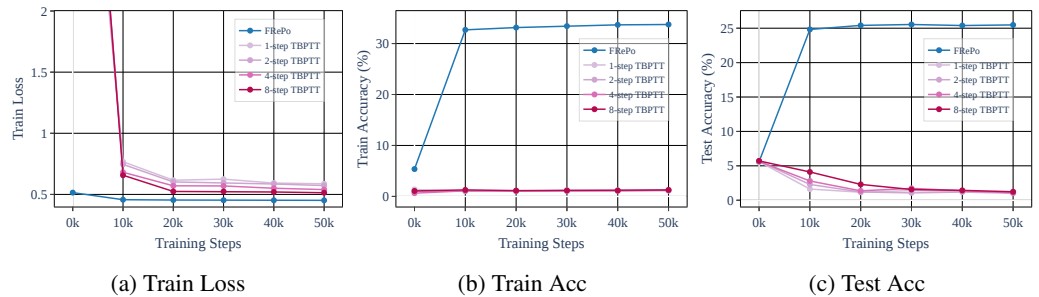

| (a) Train Loss | (b) Train Acc | (c) Test Acc |

Figure 8: Ablation Study - Truncated Backpropagation through Time (TBPTT). Due to truncation bias, k-step TBPTT can easily overfit to the k-step training scheme. In contrast, FRePo alleviates the truncation bias of TBPTT by training a subset of a neural network to convergence.

## C.2 Model pool and Batchsize

We investigate how the number of online models, the maximum online updates, and batch size affect the test accuracy. When one hyperparameter is modified, all the other hyperparameters are unchanged at their default value. As shown in Figure 9a, using more than one model is better than using only one model, which is equivalent to the training and resetting strategy used in the previous methods [5, 7, 13]. Our default choice of using ten models seems not to be the best but gives reasonable performance. Figure 9b shows that both using too few updates and using too many updates hurt the performance, and our default choice of 100 gives a good starting point. If we want to squeeze the performance, it is worth tuning these two hyperparameters. Intuitively, a small regularization strength is needed when we distill a small number of distilled data. On the contrary, when we distill more distilled data, we may want to use a stronger regularization. Figure 9c, 9d show that using a large batch size may give slightly better results and converge faster in terms of number of training steps. However, time per step is also an increasing function of batch size, so a large batch size may not give the best test accuracy and efficiency trade-off. Our default choice of 1024 may not be optimal, but it is a good starting point.

## C.3 Initialization

We initialize the distilled image using real images and initialize the distilled label using a mean-centered one-hot vector scaled by $1/(\sqrt{C/10})$ as default. Is this real initialization that explains why our distilled images look real and natural? Does random initialization give a similar result? How does the label initialization affect the performance? We answer these questions by trying different initialization schemes. We investigate initializing the distilled image from random Gaussian noise or a combination of real images and random noises. We also vary the scaling factor of the label initialization and the noise scale to figure out the best setting for dataset distillation. Figure 10a shows that initializing from random noise is not a problem and the best initialization scheme is the

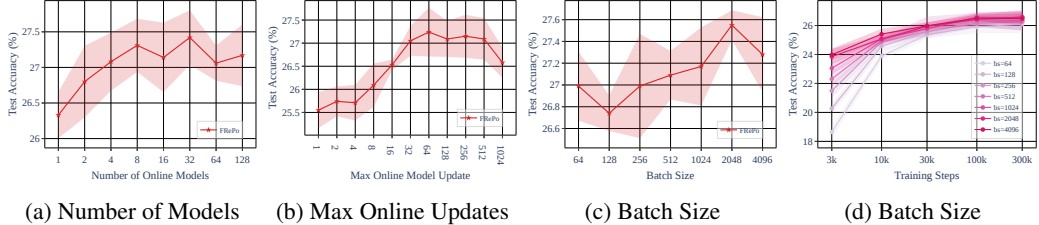

| (a) Number of Models | (b) Max Online Updates | (c) Batch Size | (d) Batch Size |

Figure 9: Ablation Study - Model Pool and Batch Size. Our default value for number of online models, max online updates, and batch size may not yield the best performance for all datasets and settings. It provides a good starting point for further investigation.

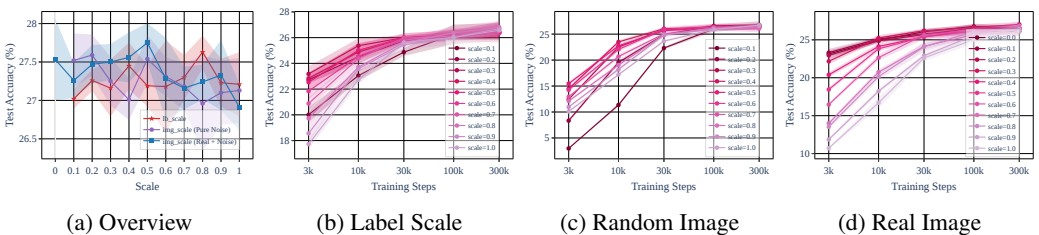

| (a) Overview | (b) Label Scale | (c) Random Image | (d) Real Image |

Figure 10: Ablation Study - Initialization. Initializing the distilled image using real images does not explain the effectiveness of our algorithm. Indeed, initializing using the combination of real image and a properly chosen random Gaussian noise gives the best performance. The scale of the label and random Gaussian noise is very crucial for convergence speed.

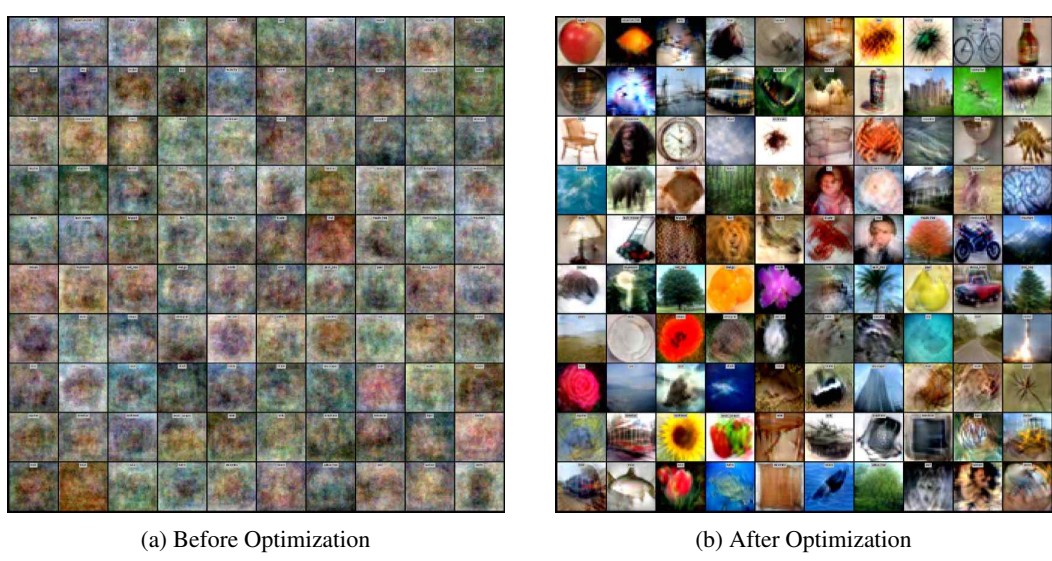

| (a) Before Optimization | (b) After Optimization |

Figure 11: Ablation Study - Initialize the distilled image using random Gaussian noise.

combination of real images and random noises. With a properly chosen standard deviation of 0.5, we can achieve 27.8% test accuracy compared to 27.2% using the default hyperparameter. Figure 10b, 10c,10d show that a right scale of noise or label can significantly improve the convergence speed. Besides, Figure 10b shows that scaling the mean-centered one-hot vector by 0.3 is a good choice for CIFAR100, and we also find that 1.0 works well for datasets with ten classes. Therefore, we decide to use the mean-centered one-hot vector scaled by $1/(\sqrt{C/10})$ as our default label initialization. As shown in Figure 11, 12, though the distilled images look very different at initialization, they look quite similar after optimization. We also provide four videos to visualize the evolution of the distilled images in the supplementary.

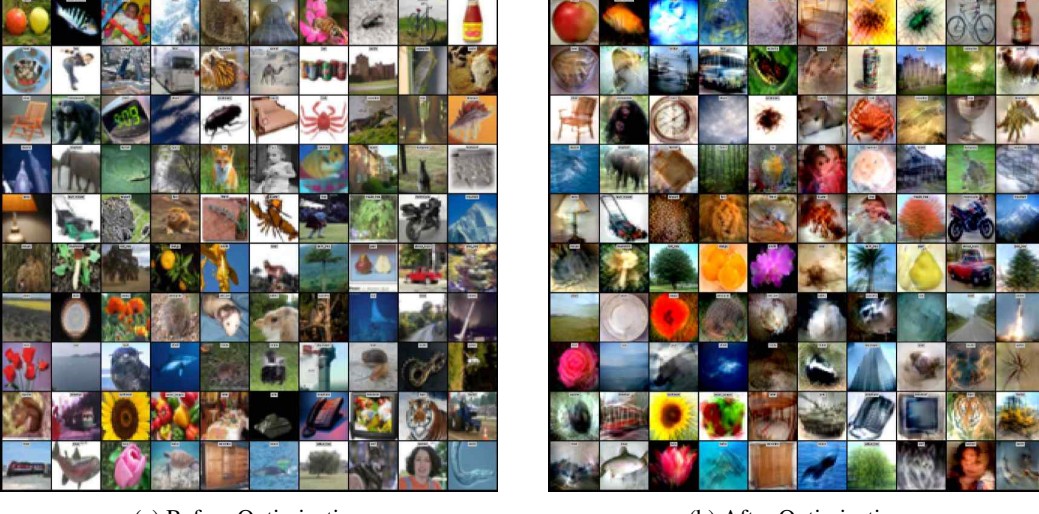

(a) Before Optimization        (b) After Optimization

Figure 12: Ablation Study - Initialize the distilled image using Real images.

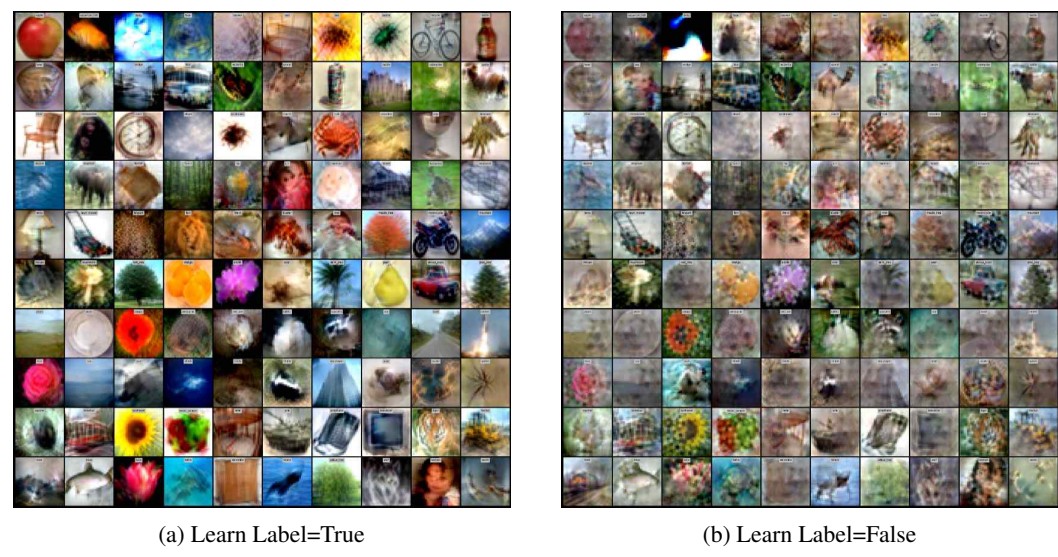

(a) Learn Label=True        (b) Learn Label=False

Figure 13: Ablation Study - Label Learning. Learning label can generate more natural and real-looking images.

## C.4 Label Learning

As discussed in Section 4.2, label learning is an essential component of our algorithm. We provide a detailed comparison of learning and not learning labels in Table 7. We observe that when the label space is simple, such as MNIST, F-MNIST, and CIFAR10, label learning may not be necessary (Figure 21, 22, 23). However, it becomes crucial for complex datasets with many classes, such as CIFAR100 (Figure 13), Tiny-ImageNet (Figure 26), and ImageNet (Figure 27, 28). For ImageNet, we can achieve 7.5% test accuracy when we learn the label, compared to 1.6% when we fix the label. Besides the fact that the distilled label encodes rich information regarding the class similarity, we find that learning labels can make the optimization easier and distill more natural and real-looking images, as shown in Figure 13, 26 and supplementary videos.

Table 7: Test accuracies of models trained on the distilled data from scratch. We highlight the best test accuracy using neural network predictor either learn the label or not.

| | | Learn Label = True | | Learn Label = False | |
|---|---|---|---|---|---|
| | Img/Cls | NN Acc | KRR Acc | NN Acc | KRR Acc |
| MNIST | 1 | $92.5 \pm 0.2$ | $92.6 \pm 0.3$ | $\mathbf{93.0 \pm 0.4}$ | $92.6 \pm 0.4$ |
| | 10 | $\mathbf{98.6 \pm 0.1}$ | $98.6 \pm 0.1$ | $\mathbf{98.6 \pm 0.1}$ | $98.6 \pm 0.1$ |
| | 50 | $\mathbf{99.2 \pm 0.0}$ | $99.2 \pm 0.1$ | $\mathbf{99.2 \pm 0.0}$ | $99.2 \pm 0.0$ |
| F-MNIST | 1 | $74.2 \pm 0.5$ | $76.4 \pm 0.3$ | $\mathbf{75.6 \pm 0.2}$ | $77.1 \pm 0.2$ |
| | 10 | $\mathbf{86.2 \pm 0.1}$ | $86.8 \pm 0.1$ | $86.0 \pm 0.1$ | $86.6 \pm 0.1$ |
| | 50 | $89.4 \pm 0.1$ | $89.9 \pm 0.1$ | $\mathbf{89.6 \pm 0.1}$ | $89.9 \pm 0.1$ |
| CIFAR10 | 1 | $45.5 \pm 0.9$ | $46.3 \pm 0.7$ | $\mathbf{46.8 \pm 0.7}$ | $47.9 \pm 0.6$ |
| | 10 | $\mathbf{65.5 \pm 0.6}$ | $68.0 \pm 0.2$ | $65.4 \pm 0.6$ | $66.9 \pm 0.4$ |
| | 50 | $\mathbf{71.7 \pm 0.2}$ | $74.4 \pm 0.1$ | $\mathbf{71.7 \pm 0.2}$ | $73.8 \pm 0.2$ |
| CIFAR100 | 1 | $\mathbf{28.7 \pm 0.1}$ | $32.3 \pm 0.1$ | $25.4 \pm 0.1$ | $27.3 \pm 0.1$ |
| | 10 | $\mathbf{42.5 \pm 0.2}$ | $44.9 \pm 0.2$ | $39.6 \pm 0.3$ | $41.5 \pm 0.1$ |
| | 50 | $\mathbf{44.3 \pm 0.2}$ | $43.0 \pm 0.3$ | $40.1 \pm 0.2$ | $37.0 \pm 0.2$ |
| T-ImageNet | 1 | $\mathbf{15.4 \pm 0.1}$ | $19.1 \pm 0.3$ | $12.4 \pm 0.8$ | $15.8 \pm 0.3$ |
| | 10 | $\mathbf{25.4 \pm 0.2}$ | $26.5 \pm 0.1$ | $21.9 \pm 0.2$ | $22.5 \pm 0.2$ |
| CUB-200 | 1 | $\mathbf{12.4 \pm 0.2}$ | $13.7 \pm 0.2$ | $7.8 \pm 0.1$ | $9.0 \pm 0.3$ |
| | 10 | $\mathbf{16.8 \pm 0.1}$ | $16.1 \pm 0.3$ | $4.8 \pm 0.4$ | $1.0 \pm 0.2$ |
| ImageNette | 1 | $\mathbf{48.1 \pm 0.7}$ | $50.6 \pm 0.6$ | $43.7 \pm 0.9$ | $46.8 \pm 0.7$ |
| | 10 | $\mathbf{66.5 \pm 0.8}$ | $67.1 \pm 0.7$ | $64.1 \pm 0.8$ | $65.1 \pm 0.8$ |
| ImageWoof | 1 | $26.7 \pm 0.6$ | $31.3 \pm 0.9$ | $\mathbf{29.7 \pm 0.6}$ | $28.2 \pm 0.9$ |
| | 10 | $\mathbf{42.2 \pm 0.9}$ | $43.5 \pm 0.8$ | $41.7 \pm 0.8$ | $43.3 \pm 0.7$ |
| ImageNet | 1 | $\mathbf{7.5 \pm 0.3}$ | $7.2 \pm 0.2$ | $1.6 \pm 0.3$ | $1.1 \pm 0.3$ |
| | 2 | $\mathbf{9.7 \pm 0.2}$ | $9.5 \pm 0.2$ | $2.0 \pm 0.3$ | $1.7 \pm 0.2$ |

## C.5 Training Cost Analysis

Similar to Figure 3, we also investigate how time per step and GPU memory usage vary when we increase the number of distilled data. As shown in Figure 14, our method becomes more expensive as we increase the number of distilled data. It is because 1) we always use all the distilled data for gradient computation rather than sample a batch as in other methods; 2) Matrix inversion with time complexity of $O(N^3)$ in KRR becomes more and more expensive as we distill more data. Similar to other kernel methods, distilling tens of thousands of data can be difficult for our method. We can circumvent this problem by 1) sampling a batch at each gradient computation, 2) performing subset distillation, or 3) distillation by class as in Section 5.1. However, the performance is expected to drop because the redundant information can be generated in different groups, and the distilled data may not be able to capture all the distinguishable features when learning from a subset. We leave it for future work to address this scaling challenge. We provide the numerical values for Figure 3c, 3d, 14a, and 14b in Table 8, 9, 10, and 11.

Table 8: Time per step measure in milliseconds (ms). Corresponds to Figure 3c.

| Width | DSA | DM | MTT | FRePo |
|---|---|---|---|---|
| 16 | $1764.3 \pm 71.2$ | $569.1 \pm 13.2$ | $206.2 \pm 2.5$ | $7.7 \pm 0.3$ |
| 32 | $1742.2 \pm 17.1$ | $573.5 \pm 11.2$ | $268.6 \pm 2.0$ | $6.9 \pm 0.2$ |
| 64 | $2014.6 \pm 15.0$ | $669.8 \pm 10.1$ | $299.9 \pm 2.1$ | $8.5 \pm 0.2$ |
| 128 | $2583.5 \pm 14.7$ | $950.6 \pm 13.7$ | $485.6 \pm 3.0$ | $10.9 \pm 0.1$ |
| 256 | $4909.8 \pm 16.9$ | $1569.2 \pm 8.5$ | $939.6 \pm 10.5$ | $20.8 \pm 0.1$ |
| 512 | $8764.6 \pm 13.7$ | $3209.5 \pm 8.6$ | $2153.6 \pm 9.0$ | $52.5 \pm 0.1$ |

Table 9: Peak GPU memory usage measured in gigabytes (GB). Corresponds to Figure 3d.

| Width | DSA | DM | MTT | FRePo |
|---|---|---|---|---|
| 16 | 0.800 | 0.678 | 0.650 | 0.036 |
| 32 | 0.986 | 0.714 | 1.260 | 0.086 |
| 64 | 1.488 | 0.888 | 2.462 | 0.178 |
| 128 | 2.014 | 1.164 | 4.858 | 0.387 |
| 256 | 4.594 | 1.718 | 9.790 | 0.814 |
| 512 | 10.768 | 2.970 | 19.110 | 2.080 |

Table 10: Time per step measure in millisecond (ms). Corresponds to Figure 14a.

| Number of Distilled Data | DSA | DM | MTT | FRePo |
|---|---|---|---|---|
| 100 | $2051.8 \pm 52.2$ | $649.4 \pm 11.7$ | $302.8 \pm 1.8$ | $10.9 \pm 0.1$ |
| 200 | $2073.4 \pm 55.4$ | $662.9 \pm 15.7$ | $575.0 \pm 7.3$ | $9.6 \pm 0.1$ |
| 400 | $1928.6 \pm 13.6$ | $681.4 \pm 10.7$ | $1077.5 \pm 8.8$ | $16.2 \pm 0.2$ |
| 800 | $1952.5 \pm 12.3$ | $722.7 \pm 18.2$ | $2169.1 \pm 12.5$ | $23.4 \pm 0.2$ |
| 1600 | $1977.4 \pm 16.4$ | $747.2 \pm 13.7$ | - | $35.3 \pm 0.2$ |
| 3200 | $2233.8 \pm 8.8$ | $727.3 \pm 18.8$ | - | $59.6 \pm 0.1$ |
| 6400 | $2467.8 \pm 8.8$ | $874.2 \pm 25.7$ | - | $123.2 \pm 0.1$ |

Table 11: Peak GPU memory usage measured in gigabytes (GB). Corresponds to Figure 14b.

| Number of Distilled Data | DSA | DM | MTT | FRePo |
|---|---|---|---|---|
| 100 | 1.488 | 0.888 | 2.464 | 0.178 |
| 200 | 1.644 | 0.964 | 4.848 | 0.308 |
| 400 | 1.916 | 1.116 | 9.662 | 0.563 |
| 800 | 2.558 | 1.464 | 19.700 | 0.721 |
| 1600 | 3.776 | 2.076 | - | 1.210 |
| 3200 | 6.056 | 3.412 | - | 2.380 |
| 6400 | 10.482 | 6.060 | - | 4.740 |

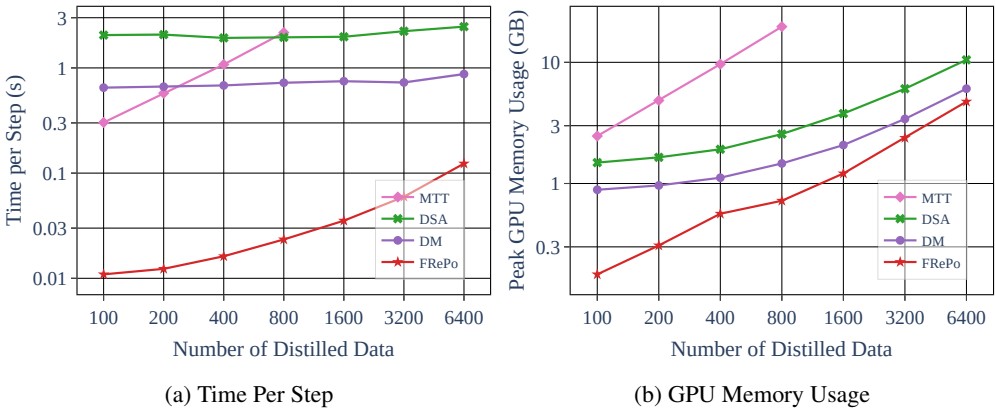

(a) Time Per Step  (b) GPU Memory Usage

Figure 14: Time per iteration and peak memory usage as we increase the number of distilled data.

## C.6 Model Architectures

From the Cross-Architecture Generalization experiments in Section 4.2, we observe that the distilled data can encode the architecture's inductive bias. Thus, we perform a qualitative and quantitative comparison of the distilled data generated by different architectures to understand how architecture affects the distilled data. We evaluate across various architectures, including Conv (Our default model), DCConv (the default model of DSA [7], DM [8], MTT [20]), AlexNet [38], VGG [39], and ResNet [40]. We also consider a wide range range of normalization layers, such as no normalization (NN), Instance Normalization (IN) [41], Batch Normalization (BN) [42], Layer Normalization (LN) [68], and Group Normalization (GN) [69]. Besides, we also vary the depth of Conv and denote two-layer, three-layer, and four-layer Conv as Conv-BN-D2, Conv-BN-D3, and Conv-BN-D4, respectively. We do not rescale any image for better visualization, so the over-saturated images indicate that the images have different statistics from the real images.

**Qualitative Results**: Figure 15, 16, 17, 18, and 19 show that 1) the simplest architecture (i.e., Conv) gives the images that look almost like real images; 2) Different normalization layers have different effects on the distilled images, resulting in images with very different brightness and contrasts. No Normalization or Batch Normalization seems to generate the most natural-looking images; 3) The distilled images generated by modern architectures like ResNet and VGG are very different from the natural images. Thus additional attention is needed to transfer the distilled images to a different architecture. On the other hand, additional tricks such as image regularization or projection may help make the distilled images more similar to natural ones. 4) Number of average pooling layers (or size of the final activation map) can affect the distilled image quality. Using a fewer pooling (i.e., Conv-BN-D2) will generate blur images or repeated objects in the images, and we find it is more obvious in high resolution such as Tiny-ImageNet. 5) Distilled images may reflect the similarity of the architecture. For instance, the images generated by Conv and DCConv are almost identical since the only architectural difference is the filter width. Besides, we observe that the images generated by AlexNet are very similar to those generated by Conv, which suggests that the inductive bias of those two architectures is very similar. It may be one reason why Conv's distilled images work extremely well for AlexNet (Table 2).

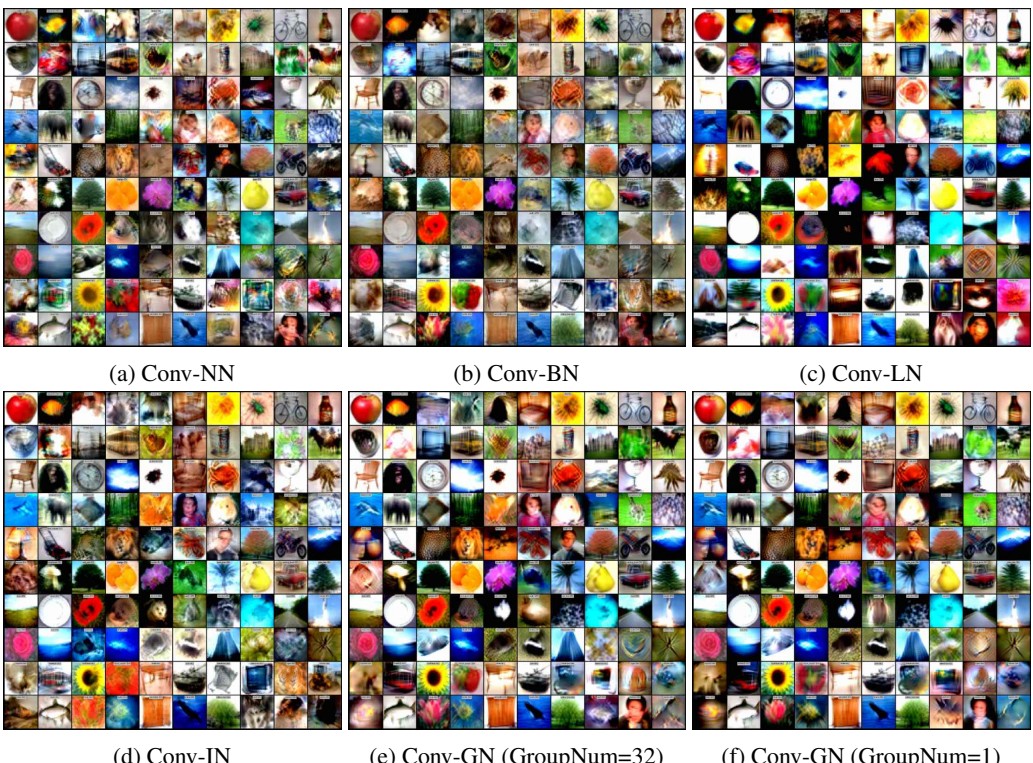

|              |              |                        |
|:------------:|:------------:|:----------------------:|
| (a) Conv-NN  | (b) Conv-BN  | (c) Conv-LN            |
| (d) Conv-IN  | (e) Conv-GN (GroupNum=32) | (f) Conv-GN (GroupNum=1) |

Figure 15: Ablation Study - Default Model with different Normalization Layer

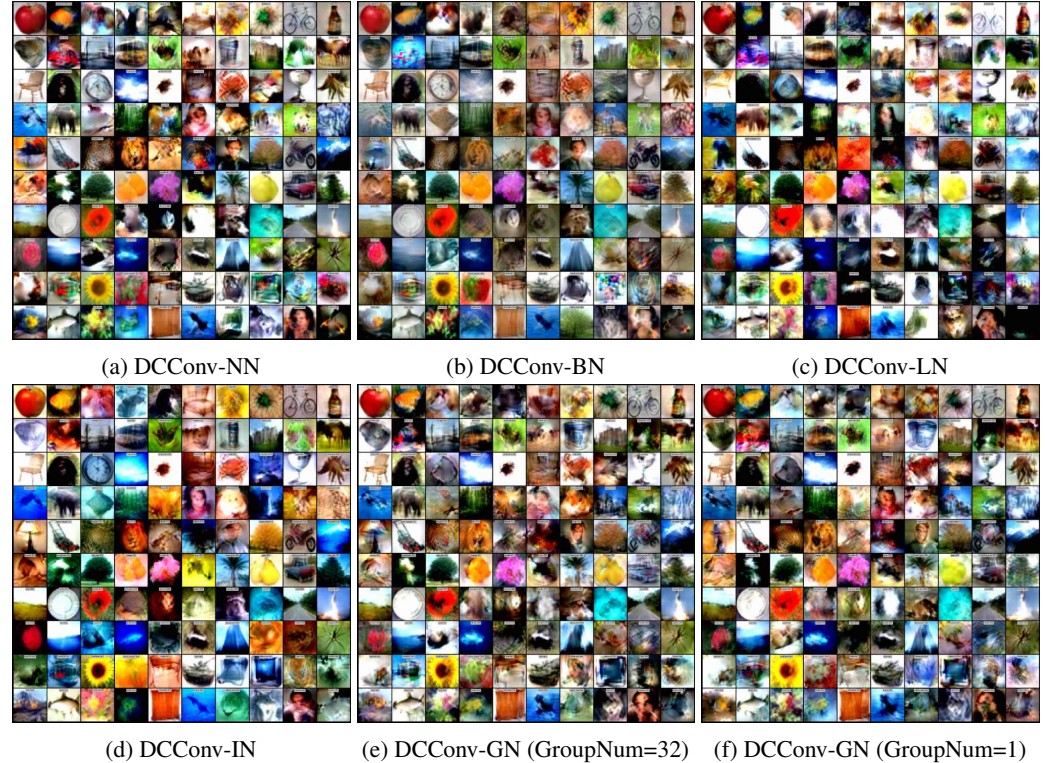

(a) DCConv-NN        (b) DCConv-BN        (c) DCConv-LN

(d) DCConv-IN      (e) DCConv-GN (GroupNum=32)      (f) DCConv-GN (GroupNum=1)

Figure 16: Ablation Study - DCConv with different Normalization Layer

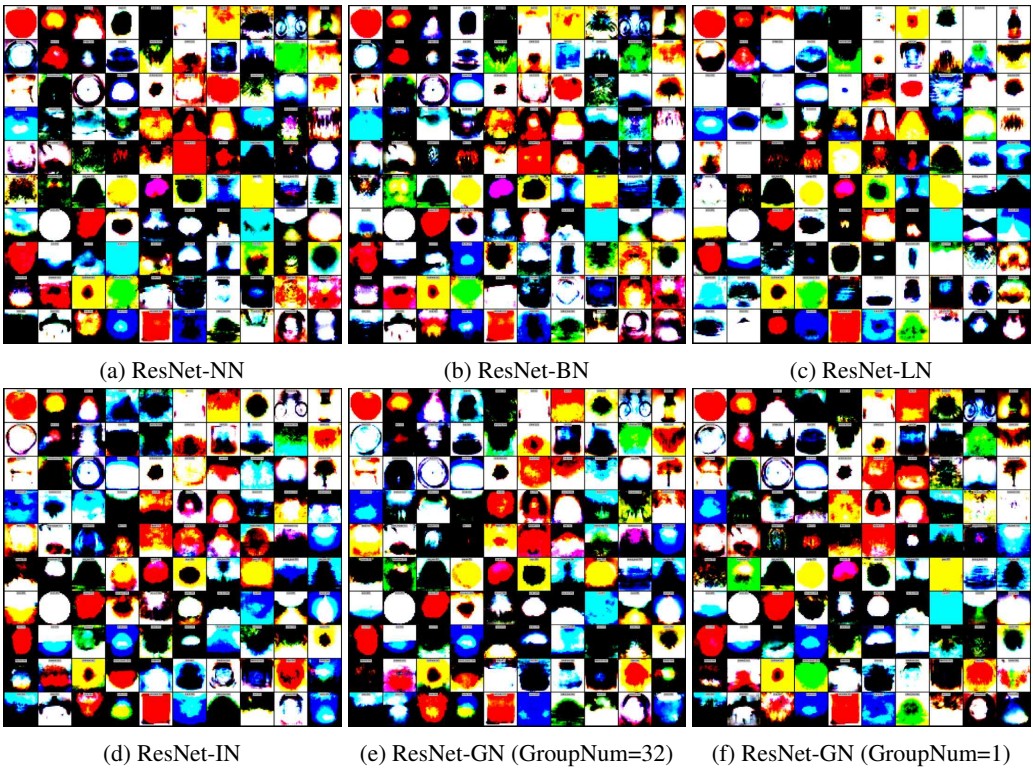

(a) ResNet-NN        (b) ResNet-BN        (c) ResNet-LN

(d) ResNet-IN      (e) ResNet-GN (GroupNum=32)      (f) ResNet-GN (GroupNum=1)

Figure 17: Ablation Study - ResNet with different Normalization Layer

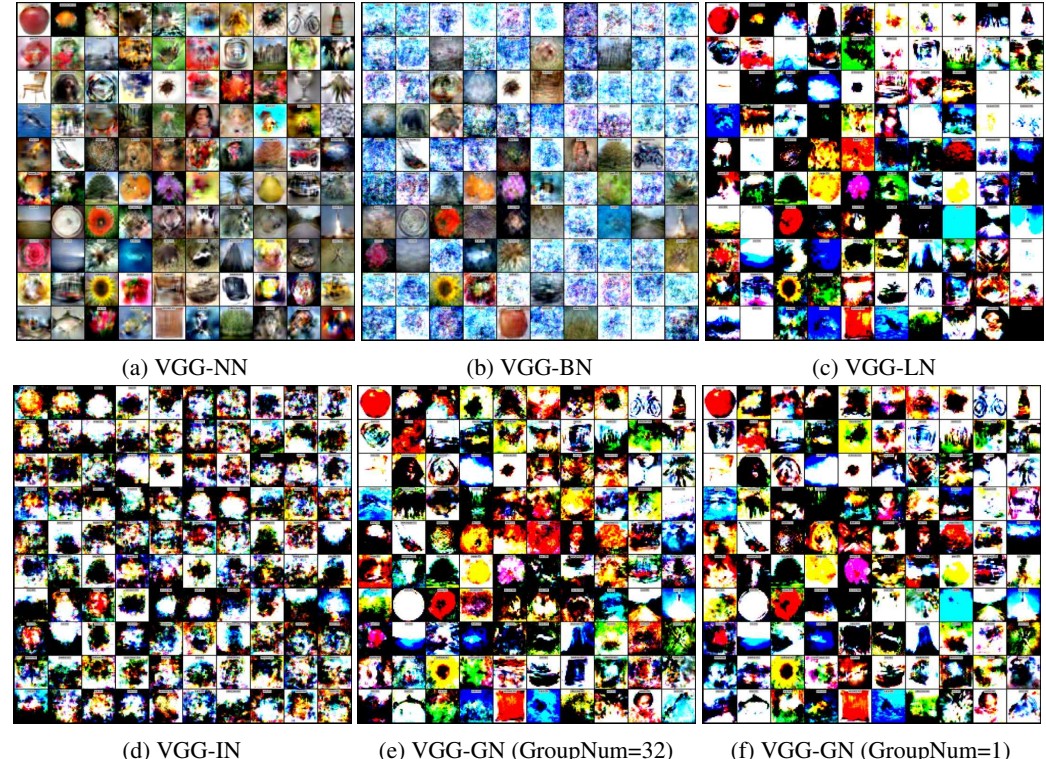

|   |   |   |
|---|---|---|
| (a) VGG-NN | (b) VGG-BN | (c) VGG-LN |
| (d) VGG-IN | (e) VGG-GN (GroupNum=32) | (f) VGG-GN (GroupNum=1) |

Figure 18: Ablation Study - VGG with different Normalization Layer.

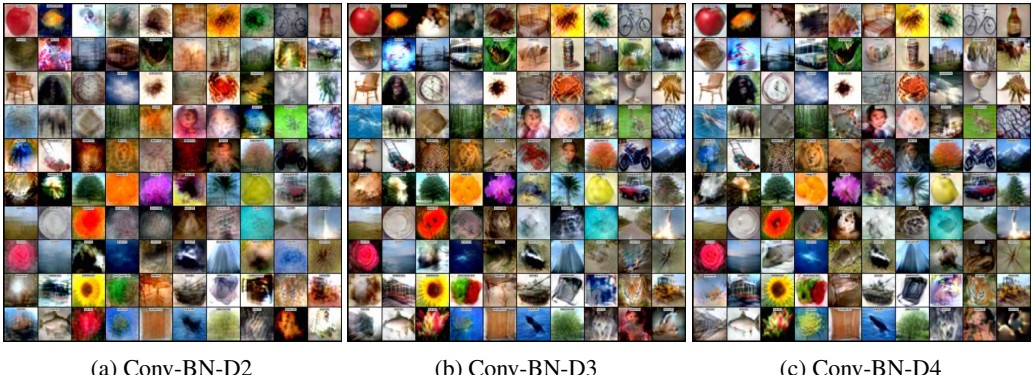

|   |   |   |
|---|---|---|
| (a) Conv-BN-D2 | (b) Conv-BN-D3 | (c) Conv-BN-D4 |

Figure 19: Ablation Study - Conv with different Depth.

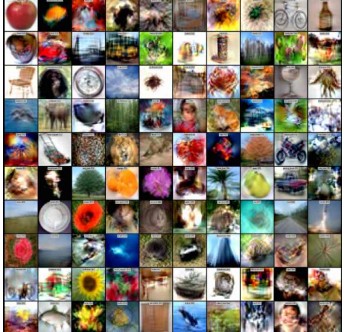

Figure 20: Ablation Study - AlexNet.

**Quantitative Results**: We evaluate the distilled data trained using different architectures (Figure 15 and 16) to study the role of normalization and width in training and evaluation. Table 12, 13, and 14 show that 1) Training with Conv-BN and evaluating using Conv-NN yields the best performance, which is our default choice; 2) Training with Conv-BN yields most generalizable and transferable images as it performs well for other architectures; 3) Evaluating using Conv-GN seems to be the best choice if the training architecture is unknown. 4) Evaluating using Conv-NN is a good way to see whether the inductive bias of architecture has been distilled to the dataset since it is very sensitive to the training architecture. There, it is not a good idea to use this architecture when the training architecture is unknown; 5) Training the distilled data using a wider network Conv achieves slightly better performance than a narrower network DCConv.

We also perform the same experiments in Table 1 using Conv-IN, and DCConv-IN and summarize the results in Table 16 and 17. Moreover, we also evaluate the cross-architecture transfer performance of Conv-IN in Table 15. We observe that DCConv-IN works reasonably well when we distill a small number of images ( 100). The performance degrades a lot when distilling 1000 images from CIFAR100 because the KRR component needs a larger feature dimension to perform well. Besides, Conv-IN performs slightly worse than the default Conv-BN. However, Table 15 suggests that the drawback of the instance norm is the transferability. The distilled data trained using instance normalization transfer less well to other architecture, especially those without normalization.

Table 12: Test accuracies of models trained on the distilled data from scratch. Each row has the same training architecture, while each column has the same evaluation architecture. We report the average performance of four random seeds. C denotes Conv and D denotes DCConv. The standard deviation are shown in Table 13, and 14. We visualize the distilled images in Figure 15 and 16.

| T\E | C-NN | C-BN | C-GN | C-GN1 | C-LN | C-IN | D-NN | D-BN | D-GN | D-GN1 | D-LN | D-IN |
|-----|------|------|------|-------|------|------|------|------|------|-------|------|------|
| C-NN | 25.7 | 22.5 | 26.5 | 26.2 | 25.3 | 23.6 | 19.1 | 21.5 | 23.8 | 24.3 | 22.0 | 22.2 |
| C-BN | 28.8 | 26.0 | 28.1 | 27.6 | 27.4 | 27.0 | 22.7 | 24.6 | 24.3 | 25.4 | 23.4 | 24.1 |
| C-GN | 26.1 | 24.1 | 28.4 | 27.4 | 26.0 | 25.6 | 20.0 | 23.8 | 25.9 | 26.3 | 24.1 | 24.7 |
| C-GN1 | 25.9 | 23.3 | 27.3 | 26.8 | 25.8 | 24.6 | 19.8 | 23.5 | 25.3 | 25.8 | 23.4 | 24.1 |
| C-LN | 21.7 | 20.6 | 23.1 | 23.2 | 23.8 | 21.0 | 14.8 | 20.4 | 22.0 | 22.8 | 23.5 | 20.4 |
| C-IN | 25.1 | 22.8 | 26.1 | 25.7 | 25.2 | 27.0 | 20.6 | 21.5 | 24.6 | 24.2 | 23.1 | 25.7 |
| D-NN | 25.1 | 22.4 | 27.3 | 27.3 | 26.3 | 22.0 | 17.2 | 21.7 | 25.6 | 26.3 | 24.1 | 23.8 |
| D-BN | 27.4 | 23.9 | 27.8 | 27.4 | 26.7 | 25.9 | 19.8 | 22.1 | 24.9 | 25.4 | 23.2 | 24.0 |
| D-GN | 24.8 | 21.1 | 26.4 | 27.0 | 25.3 | 21.1 | 15.7 | 21.1 | 25.6 | 25.6 | 22.3 | 23.1 |
| D-GN1 | 24.8 | 21.5 | 26.4 | 27.2 | 25.7 | 21.5 | 16.3 | 21.4 | 24.4 | 25.7 | 22.9 | 22.0 |
| D-LN | 24.0 | 21.7 | 24.6 | 25.1 | 25.5 | 20.7 | 13.4 | 21.3 | 21.6 | 23.1 | 25.3 | 19.8 |
| D-IN | 23.3 | 20.2 | 24.2 | 24.0 | 23.0 | 23.4 | 16.9 | 20.1 | 22.8 | 22.5 | 21.0 | 24.8 |

Table 13: Test accuracies of Conv with different normalizations evaluated on the distilled data trained using Conv and DCConv with different normalizations. We report the mean and standard deviation of four random seeds. The distilled images are shown in Figure 15 and 16.

| T\E | Conv-NN | Conv-BN | Conv-GN | Conv-GN1 | Conv-LN | Conv-IN |
|-----|---------|---------|---------|----------|---------|---------|
| Conv-NN | $25.7 \pm 0.3$ | $22.5 \pm 0.4$ | $26.5 \pm 0.2$ | $26.2 \pm 0.1$ | $25.3 \pm 0.2$ | $23.6 \pm 0.2$ |
| Conv-BN | $28.8 \pm 0.2$ | $26.0 \pm 0.3$ | $28.1 \pm 0.5$ | $27.6 \pm 0.3$ | $27.4 \pm 0.1$ | $27.0 \pm 0.3$ |
| Conv-GN | $26.1 \pm 0.3$ | $24.1 \pm 0.1$ | $28.4 \pm 0.3$ | $27.4 \pm 0.1$ | $26.0 \pm 0.1$ | $25.6 \pm 0.3$ |
| Conv-GN1 | $25.9 \pm 0.1$ | $23.3 \pm 0.1$ | $27.3 \pm 0.3$ | $26.8 \pm 0.3$ | $25.8 \pm 0.1$ | $24.6 \pm 0.2$ |
| Conv-LN | $21.7 \pm 0.2$ | $20.6 \pm 0.1$ | $23.1 \pm 0.1$ | $23.2 \pm 0.3$ | $23.8 \pm 0.3$ | $21.0 \pm 0.1$ |
| Conv-IN | $25.1 \pm 0.3$ | $22.8 \pm 0.4$ | $26.1 \pm 0.4$ | $25.7 \pm 0.2$ | $25.2 \pm 0.3$ | $27.0 \pm 0.3$ |
| DCConv-NN | $25.1 \pm 0.2$ | $22.4 \pm 0.2$ | $27.3 \pm 0.3$ | $27.3 \pm 0.5$ | $26.3 \pm 0.2$ | $22.0 \pm 0.2$ |
| DCConv-BN | $27.4 + 0.2$ | $23.9 \pm 0.1$ | $27.8 \pm 0.4$ | $27.4 \pm 0.3$ | $26.7 \pm 0.2$ | $25.9 \pm 0.3$ |
| DCConv-GN | $24.8 \pm 0.2$ | $21.1 \pm 0.1$ | $26.4 \pm 0.3$ | $27.0 \pm 0.3$ | $25.3 \pm 0.2$ | $22.1 \pm 0.3$ |
| DCConv-GN1 | $24.8 \pm 0.4$ | $21.5 \pm 0.2$ | $26.4 \pm 0.1$ | $27.2 \pm 0.1$ | $25.7 \pm 0.4$ | $21.5 \pm 0.3$ |
| DCConv-LN | $24.0 \pm 0.1$ | $21.7 \pm 0.3$ | $24.6 \pm 0.4$ | $25.1 \pm 0.1$ | $25.5 \pm 0.2$ | $20.7 \pm 0.2$ |
| DCConv-IN | $23.3 \pm 0.2$ | $20.2 \pm 0.3$ | $24.2 \pm 0.3$ | $24.0 \pm 0.2$ | $23.0 \pm 0.4$ | $23.4 \pm 0.3$ |

Table 14: Test accuracies of DCConv with different normalizations evaluated on the distilled data trained using Conv and DCConv with different normalizations. We report the mean and standard deviation of four random seeds. The distilled images are shown in Figure 15 and 16.

| T\E | DCConv-NN | DCConv-BN | DCConv-GN | DCConv-GN1 | DCConv-LN | DCConv-IN |
|---|---|---|---|---|---|---|
| Conv-NN | 19.1 ± 0.1 | 21.5 ± 0.2 | 23.8 ± 0.1 | 24.3 ± 0.3 | 22.0 ± 0.2 | 22.2 ± 0.1 |
| Conv-BN | 22.7 ± 0.1 | 24.6 ± 0.1 | 24.3 ± 0.2 | 25.4 ± 0.2 | 23.4 ± 0.1 | 24.1 ± 0.2 |
| Conv-GN | 20.0 ± 0.7 | 23.8 ± 0.3 | 25.9 ± 0.1 | 26.3 ± 0.1 | 24.1 ± 0.2 | 24.7 ± 0.2 |
| Conv-GN1 | 19.8 ± 0.7 | 23.5 ± 0.4 | 25.3 ± 0.1 | 25.8 ± 0.2 | 23.4 ± 0.1 | 24.1 ± 0.1 |
| Conv-LN | 14.8 ± 0.8 | 20.4 ± 0.3 | 22.0 ± 0.3 | 22.8 ± 0.3 | 23.5 ± 0.4 | 20.4 ± 0.2 |
| Conv-IN | 20.6 ± 0.4 | 21.5 ± 0.2 | 24.6 ± 0.1 | 24.2 ± 0.2 | 23.1 ± 0.2 | 25.7 ± 0.2 |
| DCConv-NN | 17.2 ± 0.9 | 21.7 ± 0.1 | 25.6 ± 0.2 | 26.3 ± 0.3 | 24.1 ± 0.1 | 23.8 ± 0.2 |
| DCConv-BN | 19.8 ± 0.5 | 22.1 ± 0.3 | 24.9 ± 0.3 | 25.4 ± 0.2 | 23.2 ± 0.2 | 24.0 ± 0.3 |
| DCConv-GN | 15.7 ± 0.7 | 21.1 ± 0.2 | 25.5 ± 0.3 | 25.6 ± 0.3 | 22.3 ± 0.2 | 23.1 ± 0.1 |
| DCConv-GN1 | 16.3 ± 0.8 | 21.4 ± 0.2 | 24.4 ± 0.2 | 25.7 ± 0.4 | 22.9 ± 0.3 | 22.0 ± 0.2 |
| DCConv-LN | 13.4 ± 1.1 | 21.3 ± 0.4 | 21.6 ± 0.5 | 23.1 ± 0.3 | 25.3 ± 0.3 | 19.8 ± 0.3 |
| DCConv-IN | 16.9 ± 0.8 | 20.1 ± 0.3 | 22.8 ± 0.1 | 22.5 ± 0.2 | 21.0 ± 0.4 | 24.8 ± 0.3 |

Table 15: Cross-architecture transfer performance on CIFAR10 with 10 Img/Cls. Despite being trained for a specific architecture, our distilled data transfer well to various architectures unseen during training. Conv is the default evaluation model used for each method. NN, DN, IN, and BN stand for no normalization, default normalization, Instance Normalization, Batch Normalization, respectively.

| | Train Arch | Evaluation Architecture | | | | | | |
|---|---|---|---|---|---|---|---|---|
| | | Conv | Conv-NN | ResNet-DN | ResNet-BN | VGG-DN | VGG-BN | AlexNet |
| DSA [7] | Conv-IN | 53.2 ± 0.8 | 36.4 ± 1.5 | 42.1 ± 0.7 | 34.1 ± 1.4 | 48.3 ± 0.7 | 46.3 ± 1.3 | 34.0 ± 2.3 |
| DM [8] | Conv-IN | 49.2 ± 0.8 | 35.2 ± 0.5 | 36.8 ± 1.2 | 35.5 ± 1.3 | 45.5 ± 1.0 | 41.2 ± 1.8 | 34.9 ± 1.1 |
| MTT [20] | Conv-IN | 64.4 ± 0.9 | 41.6 ± 1.3 | 49.2 ± 1.1 | 42.9 ± 1.5 | 35.7 ± 3.35 | 46.6 ± 2.0 | 34.2 ± 2.6 |
| KIP [23] | Conv-NTK | 62.7 ± 0.3 | 58.2 ± 0.4 | 49.0 ± 1.2 | 45.8 ± 1.4 | 32.0 ± 0.4 | 30.1 ± 1.5 | 57.2 ± 0.4 |
| FRePo | Conv-IN | 59.2 ± 0.3 | 56.2 ± 0.2 | 51.1 ± 0.8 | 50.8 ± 0.2 | **57.5 ± 0.7** | 51.8 ± 0.3 | 55.3 ± 0.8 |
| FRePo | Conv-BN | **65.5 ± 0.4** | **65.5 ± 0.4** | **58.1 ± 0.6** | **57.7 ± 0.7** | 49.1 ± 0.5 | **59.4 ± 0.7** | **61.9 ± 0.7** |

Table 16: NN test accuracies of models trained on the distilled data from scratch using Conv, Conv-IN, and DCConv-IN. $^\dagger$ denotes performance better than the original reported performance.

| | Img/Cls | Previous SOTA | | | | FRePo | | |
|---|---|---|---|---|---|---|---|---|
| | | DSA [7] | DM [8] | KIP [23] | MTT [20] | Conv-BN | Conv-IN | DCConv-IN |
| MNIST | 1 | 88.7 ± 0.6 | 89.9 ± 0.8$^\dagger$ | 90.1 ± 0.1 | 91.4 ± 0.9$^\dagger$ | **93.0 ± 0.4** | 92.9 ± 0.5 | 92.4 ± 0.5 |
| | 10 | 97.9 ± 0.1$^\dagger$ | 97.6 ± 0.1$^\dagger$ | 97.5 ± 0.0 | 97.3 ± 0.1$^\dagger$ | **98.6 ± 0.1** | 98.9 ± 0.1 | 98.4 ± 0.1 |
| | 50 | 99.2 ± 0.1 | 98.6 ± 0.1 | 98.3 ± 0.1 | 98.5 ± 0.1$^\dagger$ | 99.2 ± 0.0 | **99.4 ± 0.1** | 98.8 ± 0.1 |
| F-MNIST | 1 | 70.6 ± 0.6 | 71.5 ± 0.5$^\dagger$ | 73.5 ± 0.5 | 75.1 ± 0.9$^\dagger$ | 75.6 ± 0.3 | 75.3 ± 0.8 | **76.4 ± 1.2** |
| | 10 | 84.8 ± 0.3$^\dagger$ | 83.6 ± 0.2$^\dagger$ | 86.8 ± 0.1 | **87.2 ± 0.3**$^\dagger$ | 86.2 ± 0.2 | 86.0 ± 0.3 | 85.7 ± 0.2 |
| | 50 | 88.8 ± 0.2$^\dagger$ | 88.2 ± 0.1$^\dagger$ | 88.0 ± 0.1 | 88.3 ± 0.1$^\dagger$ | **89.6 ± 0.1** | 89.4 ± 0.1 | 87.2 ± 0.2 |
| CIFAR10 | 1 | 36.7 ± 0.8$^\dagger$ | 31.0 ± 0.6$^\dagger$ | **49.9 ± 0.2** | 46.3 ± 0.8 | 46.8 ± 0.7 | 45.1 ± 0.5 | 41.3 ± 0.5 |
| | 10 | 53.2 ± 0.8$^\dagger$ | 49.2 ± 0.8$^\dagger$ | 62.7 ± 0.3 | 65.3 ± 0.7 | **65.5 ± 0.4** | 59.1 ± 0.3 | 59.6 ± 0.3 |
| | 50 | 66.8 ± 0.4$^\dagger$ | 63.7 ± 0.5$^\dagger$ | 68.6 ± 0.2 | 71.6 ± 0.2 | **71.7 ± 0.2** | 69.6 ± 0.4 | 63.6 ± 0.2 |
| CIFAR100 | 1 | 16.8 ± 0.2$^\dagger$ | 12.2 ± 0.4$^\dagger$ | 15.7 ± 0.2 | 24.3 ± 0.3 | **28.7 ± 0.1** | 25.9 ± 0.1 | 24.8 ± 0.2 |
| | 10 | 32.3 ± 0.3 | 29.7 ± 0.3 | 28.3 ± 0.1 | 40.1 ± 0.4 | **42.5 ± 0.2** | 40.9 ± 0.1 | 31.2 ± 0.1 |
| | 50 | 42.8 ± 0.4 | 43.6 ± 0.4 | — | **47.7 ± 0.2** | 44.3 ± 0.2 | — | — |
| T-ImageNet | 1 | 6.6 ± 0.2$^\dagger$ | 3.9 ± 0.2 | — | 8.8 ± 0.3 | **15.4 ± 0.1** | 13.5 ± 0.1 | — |
| | 10 | — | 12.9 ± 0.4 | — | 23.2 ± 0.2 | **25.4 ± 0.2** | 20.4 ± 0.1 | — |

Table 17: KRR test accuracies of FRePo trained on the distilled data from scratch using Conv, Conv-IN, and DCConv-IN. $^{\dagger}$ denotes performance better than the original reported performance.

| | Img/Cls | Previous SOTA | | | | FRePo | | |
|---|---|---|---|---|---|---|---|---|
| | | DSA [7] | DM [8] | KIP [23] | MTT [20] | Conv-BN | Conv-IN | DCConv-IN |
| MNIST | 1 | $88.7 \pm 0.6$ | $89.9 \pm 0.8^{\dagger}$ | $90.1 \pm 0.1$ | $91.4 \pm 0.9^{\dagger}$ | $92.6 \pm 0.4$ | $\mathbf{92.7 \pm 0.3}$ | $91.1 \pm 0.5$ |
| | 10 | $97.9 \pm 0.1^{\dagger}$ | $97.6 \pm 0.1^{\dagger}$ | $97.5 \pm 0.0$ | $97.3 \pm 0.1^{\dagger}$ | $98.6 \pm 0.1$ | $\mathbf{98.8 \pm 0.1}$ | $98.4 \pm 0.1$ |
| | 50 | $\mathbf{99.2 \pm 0.1}$ | $98.6 \pm 0.1$ | $98.3 \pm 0.1$ | $98.5 \pm 0.1^{\dagger}$ | $99.2 \pm 0.1$ | $\mathbf{99.3 \pm 0.1}$ | $98.9 \pm 0.1$ |
| F-MNIST | 1 | $70.6 \pm 0.6$ | $71.5 \pm 0.5^{\dagger}$ | $73.5 \pm 0.5$ | $75.1 \pm 0.9^{\dagger}$ | $77.1 \pm 0.2$ | $71.7 \pm 1.2$ | $\mathbf{78.5 \pm 0.2}$ |
| | 10 | $84.8 \pm 0.3^{\dagger}$ | $83.6 \pm 0.2^{\dagger}$ | $86.8 \pm 0.1$ | $\mathbf{87.2 \pm 0.3}^{\dagger}$ | $86.8 \pm 0.1$ | $86.9 \pm 0.2$ | $86.2 \pm 0.1$ |
| | 50 | $88.8 \pm 0.2^{\dagger}$ | $88.2 \pm 0.1^{\dagger}$ | $88.0 \pm 0.1$ | $88.3 \pm 0.1^{\dagger}$ | $\mathbf{89.9 \pm 0.1}$ | $89.9 \pm 0.1$ | $87.4 \pm 0.2$ |
| CIFAR10 | 1 | $36.7 \pm 0.8^{\dagger}$ | $31.0 \pm 0.6^{\dagger}$ | $\mathbf{49.9 \pm 0.2}$ | $46.3 \pm 0.8$ | $47.9 \pm 0.6$ | $46.8 \pm 0.3$ | $43.3 \pm 0.5$ |
| | 10 | $53.2 \pm 0.8^{\dagger}$ | $49.2 \pm 0.8^{\dagger}$ | $62.7 \pm 0.3$ | $65.3 \pm 0.7$ | $\mathbf{68.0 \pm 0.2}$ | $61.9 \pm 0.4$ | $61.8 \pm 0.3$ |
| | 50 | $66.8 \pm 0.4^{\dagger}$ | $63.7 \pm 0.5^{\dagger}$ | $68.6 \pm 0.2$ | $71.6 \pm 0.2$ | $\mathbf{74.4 \pm 0.1}$ | $71.4 \pm 0.3$ | $64.3 \pm 0.1$ |
| CIFAR100 | 1 | $16.8 \pm 0.2^{\dagger}$ | $12.2 \pm 0.4^{\dagger}$ | $15.7 \pm 0.2$ | $24.3 \pm 0.3$ | $\mathbf{32.3 \pm 0.1}$ | $25.7 \pm 0.2$ | $26.9 \pm 0.1$ |
| | 10 | $32.3 \pm 0.3$ | $29.7 \pm 0.3$ | $28.3 \pm 0.1$ | $40.1 \pm 0.4$ | $\mathbf{44.9 \pm 0.2}$ | $42.0 \pm 0.3$ | $30.9 \pm 0.1$ |
| | 50 | $42.8 \pm 0.4$ | $43.6 \pm 0.4$ | — | $\mathbf{47.7 \pm 0.2}$ | $43.0 \pm 0.3$ | — | — |
| T-ImageNet | 1 | $6.6 \pm 0.2^{\dagger}$ | $3.9 \pm 0.2$ | — | $8.8 \pm 0.3$ | $\mathbf{19.1 \pm 0.3}$ | $15.8 \pm 0.3$ | — |
| | 10 | — | $12.9 \pm 0.4$ | — | $23.2 \pm 0.2$ | $\mathbf{26.5 \pm 0.1}$ | $20.8 \pm 0.1$ | — |

## D   Hyperparameter Tuning Guideline

We find that several modifications to the current method can improve the test accuracy of the model trained on the distilled data. We do not include them in our current algorithm for simplicity or fair comparison, and we guess it may be helpful for practitioners.

- **Dropout:** We find dropout is very effective at alleviating the overfitting when training on a small set of examples.

- **Learning Rate Schedule for the online model:** Though we use a constant schedule in our experiments, we find that the learning rate schedule, especially the warm-up phase, may be crucial for some architectures. When NAN is in gradient, adding a learning rate schedule to the online model may solve the problem.

- **Data Augmentation during Training:** There are two ways to add data augmentation during training. One is to add to $X_s$ during the online model update. The other is to add to $X_t$, which can be thought of as distilling data augmentation to the data.

- **Tune Max Online Update:** We train the online model up to 100 steps in our experiments. When the distillation size is tiny (e.g., ten examples in total), 100 may be too large. Setting to a lower value (less regularization) turns out to be better. On the contrary, if the distillation size is large, setting it to a higher value (more regularization) can give better results.

- **Exponential Moving Average (EMA):** We find that evaluating on the EMA version of the distilled data or using the EMA version of the model parameters can improve the test accuracy.

- **Soft Cross Entropy Loss:** We train models on the distilled dataset using MSE loss in all our experiments to take advantage of the distilled label. An alternative way is to use the soft cross-entropy loss with a fine-tuned temperature, which usually outperforms the MSE loss.

- **Image Regularization:** Though our method does not have any image regularization, we find a regularization term on the image norm is necessary for some architectures. Otherwise, the image norm will keep increasing or decreasing. We find that using an L2 penalty between the distilled image norm and the real image norm is enough in some cases. An alternative way is to project the distilled image to a norm ball every few iterations.

- **Label Regularization:** Our method has a small regularization to ensure the class-balanced distillation where we force the margin between the target label and any other label is greater than 1/C. We believe a better label regularization incorporating prior knowledge of class similarity can improve the performance.

- **Maximal Update Parametrization ($\mu P$) [70]:** We observe that using a model parameterization proposed by Yang et al. [70] can give additional test accuracy improvement.

Based on our experience, we provide the following hyperparameter tuning guideline for those who want to squeeze the performance of our method or apply our method to a different dataset or use a different model. Besides validation loss and accuracy, we suggest monitoring the norm and gradient norm of the distilled images and labels, which can be good indicators for the final performance.

- **Step1 - Online Model:** Choose an online model architecture and tune its hyperparameter (e.g., optimizer, weight decay) in the standard way on the whole real dataset or a subset of real data. The same hyperparameters can be used for the online model update and final evaluation.

- **Step2 - Distilled Data Optimization and Initialization:** Use the default setting for the model pool (i.e., ten models with max online update $K = 100$) and tune the learning rate and batch size for distilled data optimization and the scale of initialization.

- **Step3 - Pool Diversity:** Tune the model pool diversity by adjusting the max online update, adding models with different architectures, or applying data augmentation.

# E  Additional Visualization

### E.1  Distilled Image Visualization

We provide some additional distilled images visualization for MNIST (Figure 21), FashionMNIST (Figure 22), CIFAR10 (Figure 23), CIFAR100 (Figure 24), CUB-200 (Figure 25), Tiny ImageNet (Figure 26), ImageNet (Figure 27, 28), ImageNette (Figure 29), and ImageWoof (Figure 30).

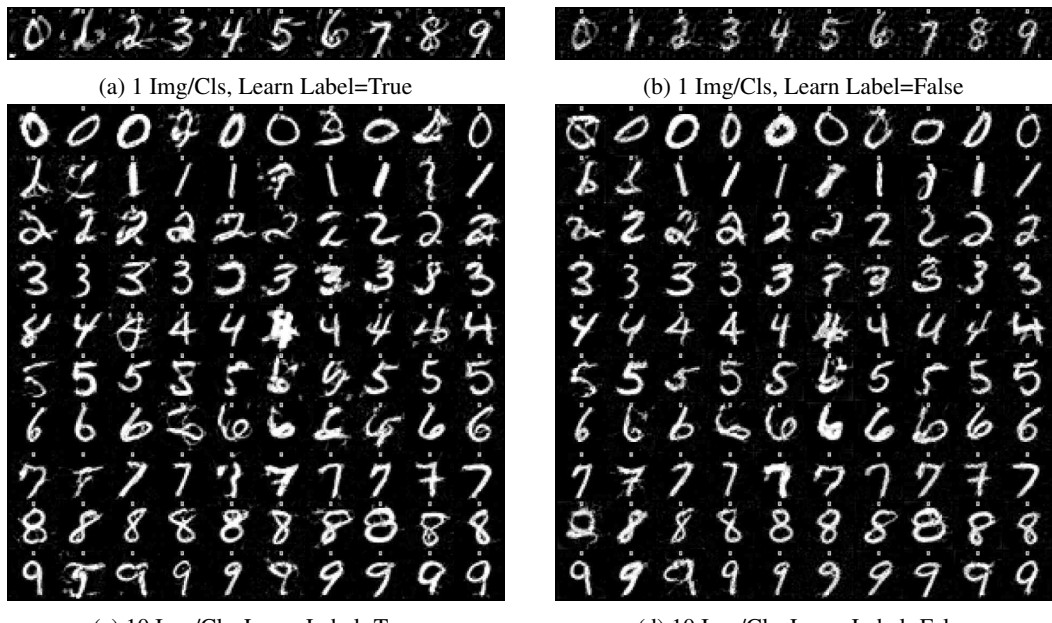

(a) 1 Img/Cls, Learn Label=True        (b) 1 Img/Cls, Learn Label=False

(c) 10 Img/Cls, Learn Label=True        (d) 10 Img/Cls, Learn Label=False

Figure 21: Distilled Image Visualization - MNIST.

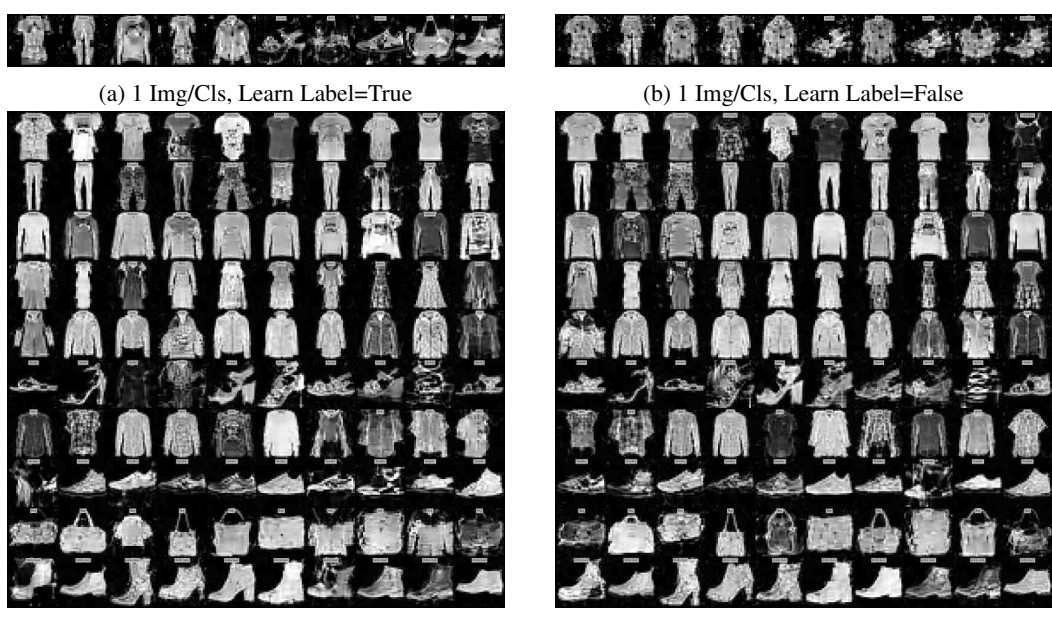

(a) 1 Img/Cls, Learn Label=True        (b) 1 Img/Cls, Learn Label=False

(c) 10 Img/Cls, Learn Label=True        (d) 10 Img/Cls, Learn Label=False

Figure 22: Distilled Image Visualization - Fashion MNIST.

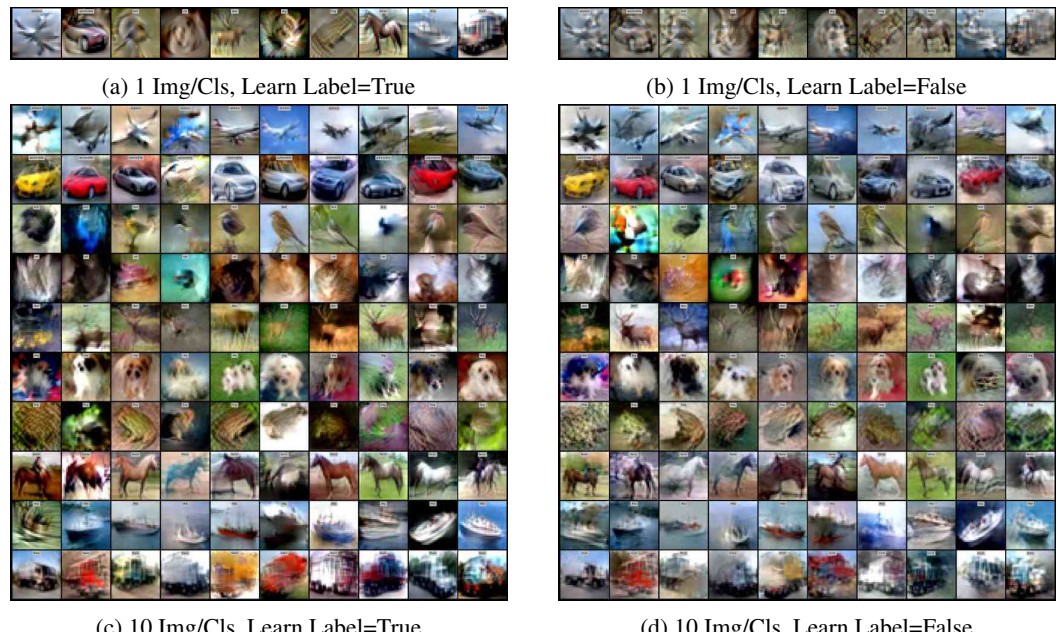

(a) 1 Img/Cls, Learn Label=True

(b) 1 Img/Cls, Learn Label=False

(c) 10 Img/Cls, Learn Label=True

(d) 10 Img/Cls, Learn Label=False

Figure 23: Distilled Image Visualization - CIFAR10

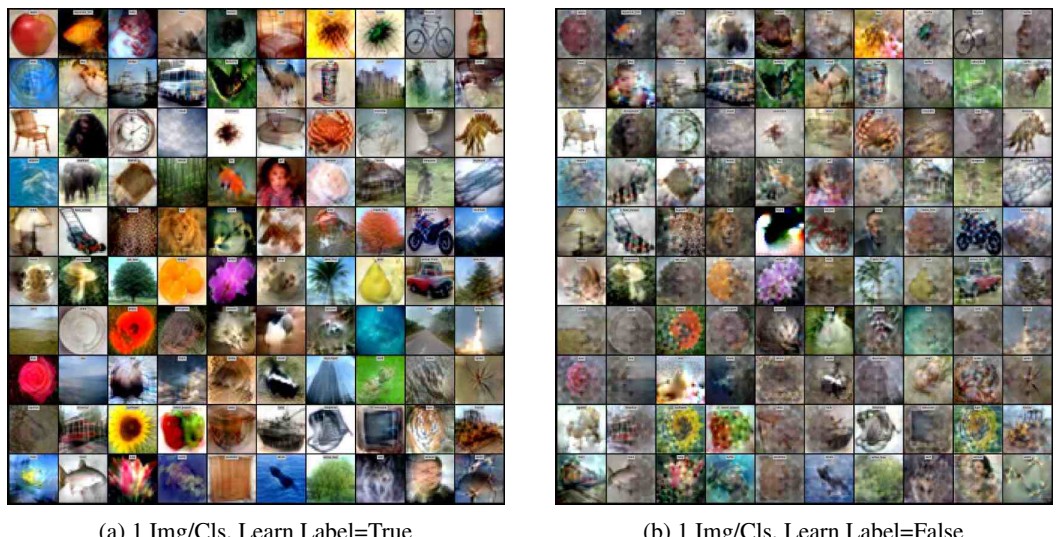

(a) 1 Img/Cls, Learn Label=True

(b) 1 Img/Cls, Learn Label=False

Figure 24: Distilled Image Visualization - CIFAR100

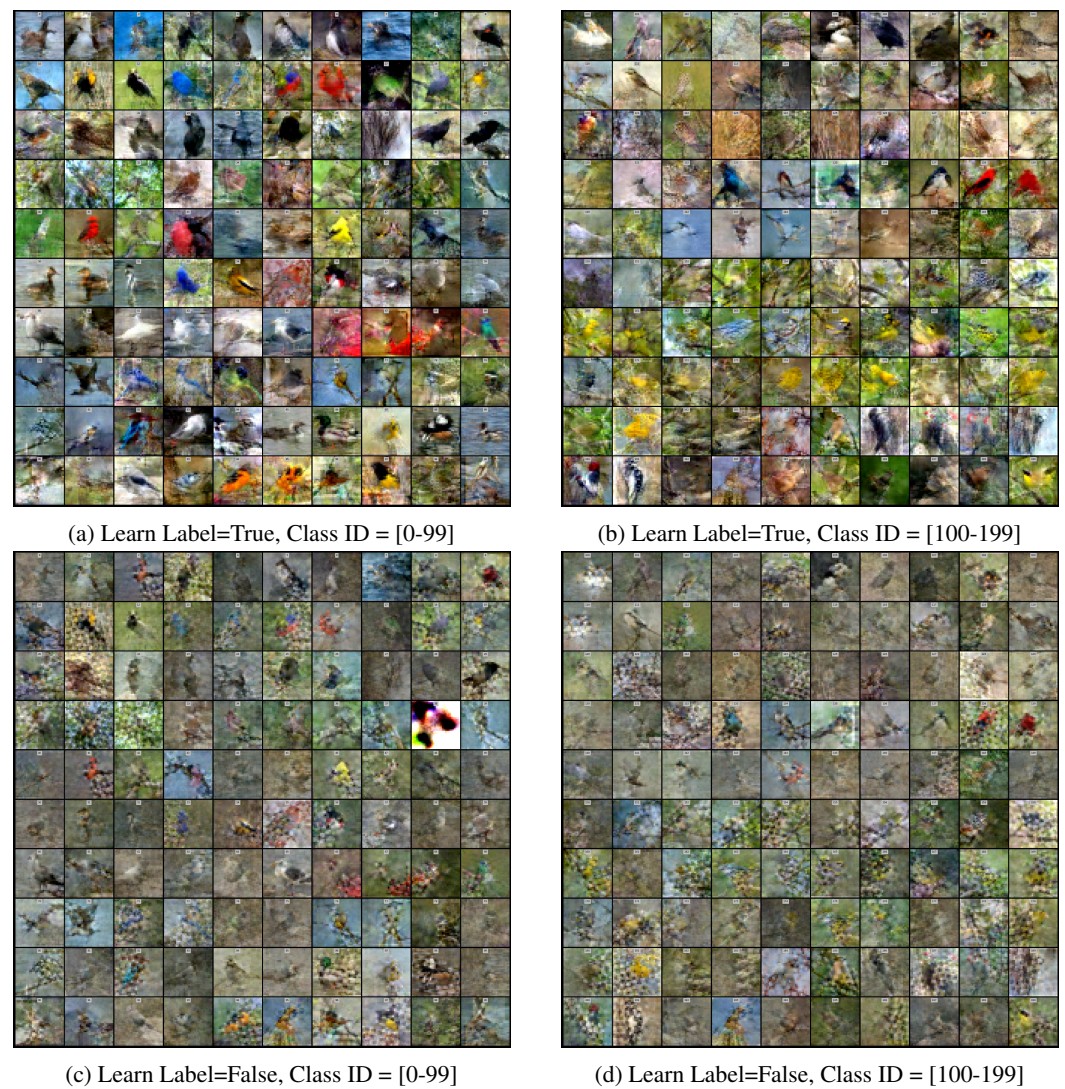

(a) Learn Label=True, Class ID = [0-99]

(b) Learn Label=True, Class ID = [100-199]

(c) Learn Label=False, Class ID = [0-99]

(d) Learn Label=False, Class ID = [100-199]

Figure 25: Distilled Image Visualization - CUB-200 (1 Img/Cls)

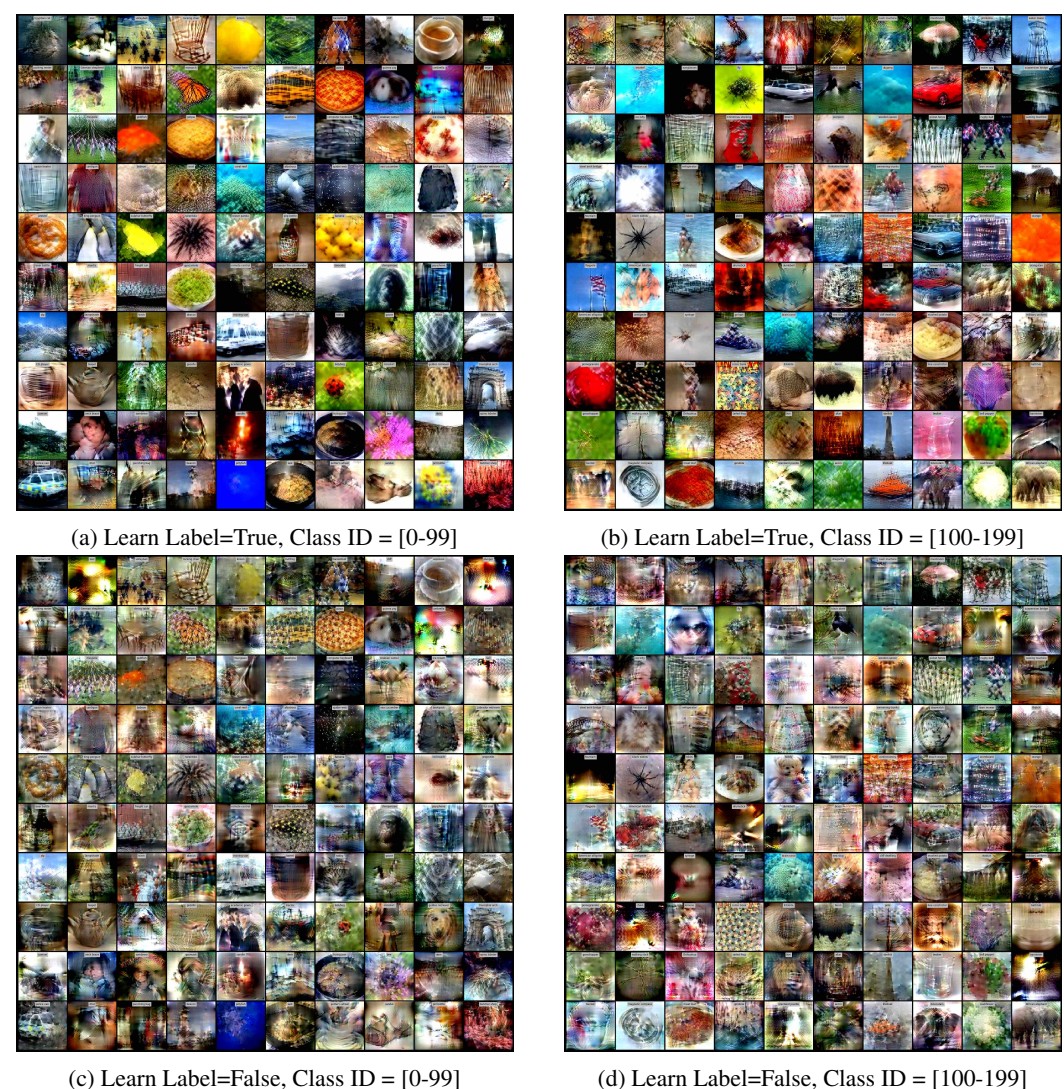

(a) Learn Label=True, Class ID = [0-99]

(b) Learn Label=True, Class ID = [100-199]

(c) Learn Label=False, Class ID = [0-99]

(d) Learn Label=False, Class ID = [100-199]

Figure 26: Distilled Image Visualization - Tiny ImageNet (1 Img/Cls)

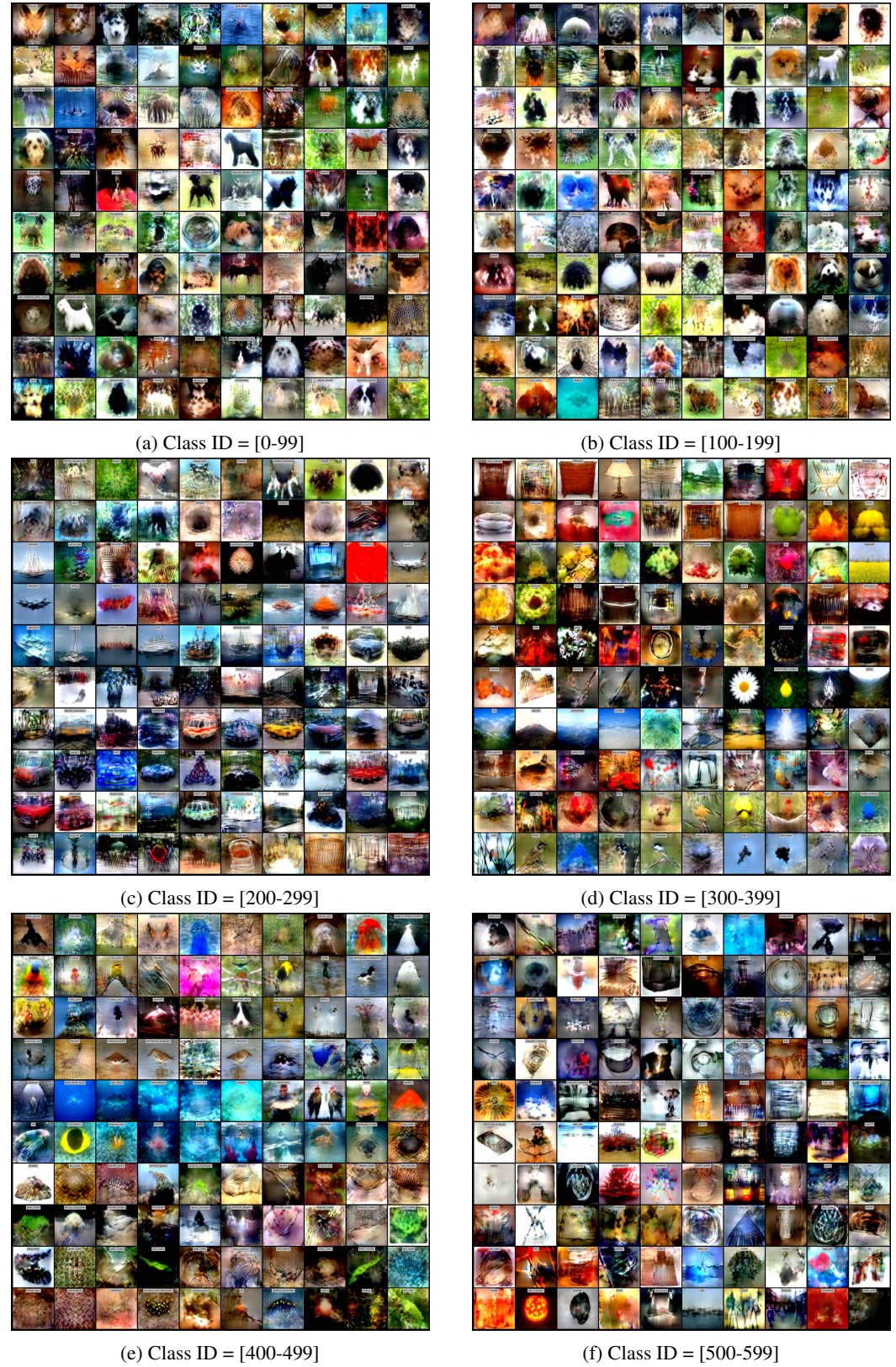

(a) Class ID = [0-99]

(b) Class ID = [100-199]

(c) Class ID = [200-299]

(d) Class ID = [300-399]

(e) Class ID = [400-499]

(f) Class ID = [500-599]

Figure 27: Distilled Image Visualization - ImageNet (1 Img/Cls), Learn Label=True

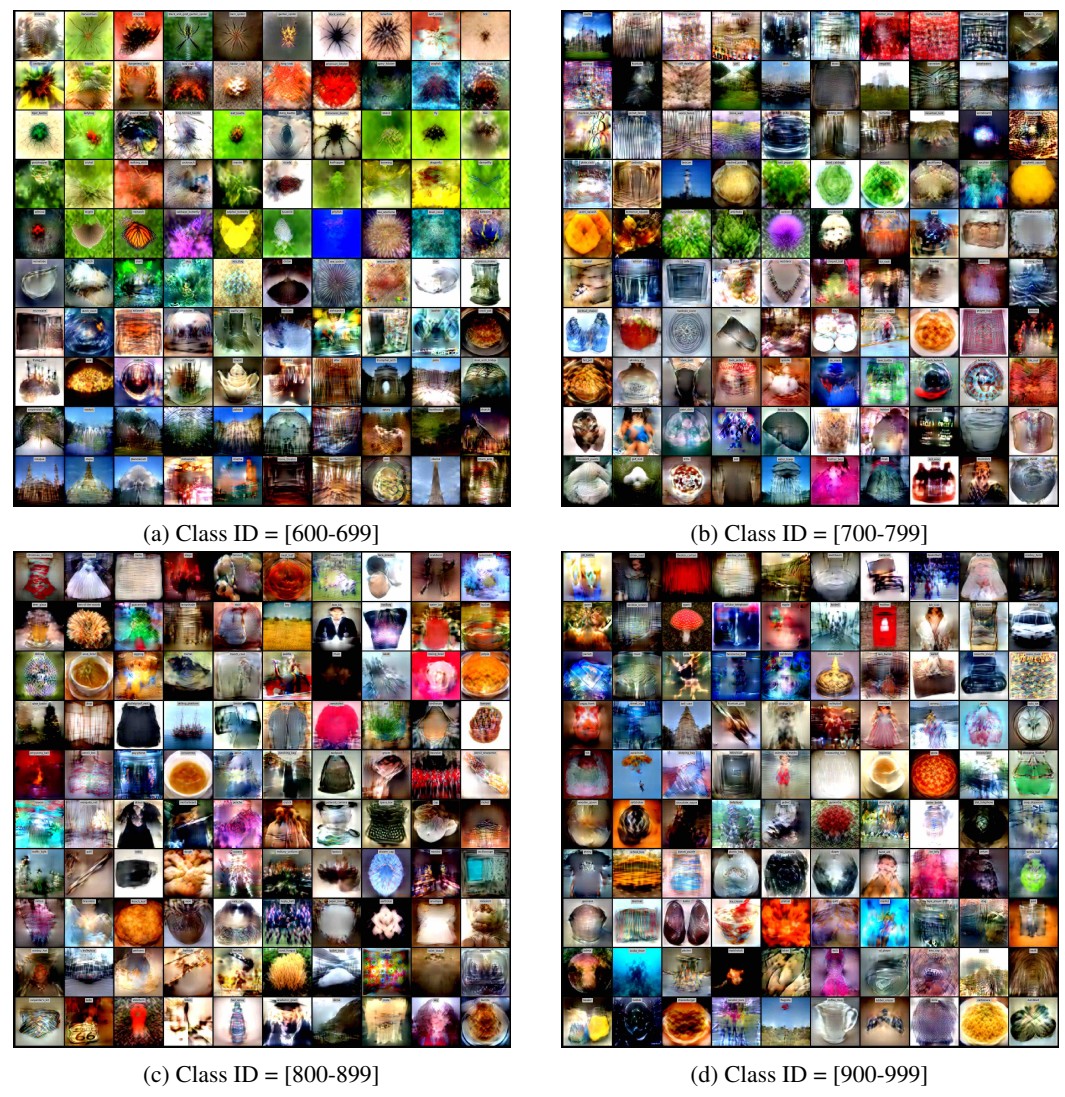

(a) Class ID = [600-699]

(b) Class ID = [700-799]

(c) Class ID = [800-899]

(d) Class ID = [900-999]

Figure 28: Distilled Image Visualization - ImageNet (1 Img/Cls), Learn Label=True

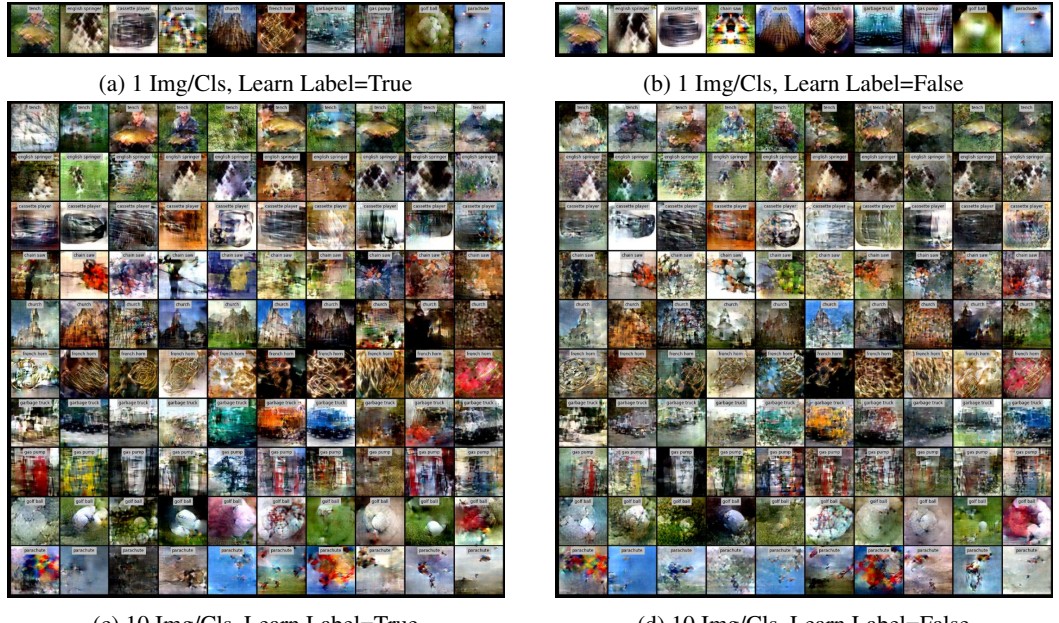

(a) 1 Img/Cls, Learn Label=True        (b) 1 Img/Cls, Learn Label=False

(c) 10 Img/Cls, Learn Label=True        (d) 10 Img/Cls, Learn Label=False

Figure 29: Distilled Image Visualization - ImageNette.

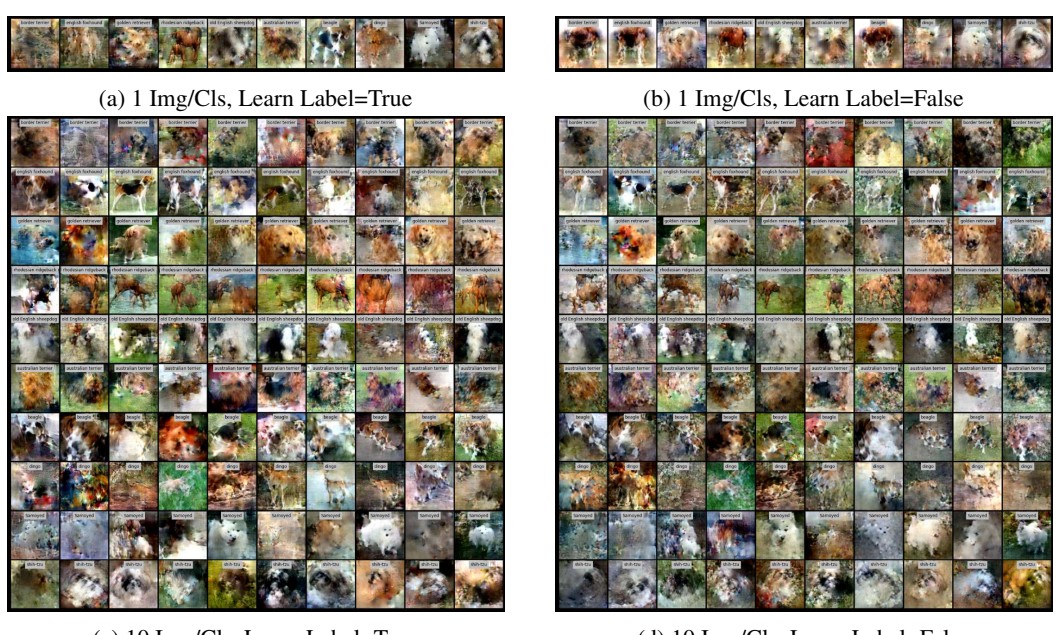

(a) 1 Img/Cls, Learn Label=True        (b) 1 Img/Cls, Learn Label=False

(c) 10 Img/Cls, Learn Label=True        (d) 10 Img/Cls, Learn Label=False

Figure 30: Distilled Image Visualization - ImageWoof.