# OpenReview forum: "Dataset Distillation using Neural Feature Regression"
_NeurIPS.cc/2022/Conference — NeurIPS 2022 Accept_

### Official Review · Reviewer_JSgu · 2022-07-04

**Rating:** 7
**Confidence:** 3
**Soundness:** 3 good
**Presentation:** 4 excellent
**Contribution:** 3 good

**Summary:**

This paper presents an efficient method for dataset distillation -- the goal is to extract/synthesize a small set of images that represent a dataset, so that deep learning models trained on the small set can achieve high (as high as possible) classification accuracy. The core contribution of this ppaer is to simplify and accelerate the conventional two-stage optimization procedure with TBPTT, where the gradients need to be propagated through a long chain. Instead, a kernel is used for computing the loss function, minimizing which leads to a single-step optimization approach which is highly efficient. The method is evaluated over a series of popular datasets including ImageNet which was never studied before.

**Questions:**

1. What is the major difficulty that prevents the proposed method from being applied to large images, e.g. 256x256 images? If one uses a small resolution setting to extract synthesized images (e.g. 64x64) and using techniques like GAN to up-sample it larger (e.g. 128x128), does it bring good performance on large datasets (e.g. ImageNet)?

2. What is the time and memory consumption for the ImageNet experiments, in particular, how long did the program run, and how many GPUs  (or other device) did it use? In addition, if one hopes to extract more training instances from ImageNet (e.g. 5, 10, 20, ... images per class), what is the computational cost w.r.t. the number of training instances?

3. If one has extracted, say, 10 images per class, and hopes to add 5 images per class for better performance, does the algorithm need to start from the very beginning (to extract 15 images per class), or can it start from the current point? In other words, what is the incremental performance of the proposed approach?

4. Are there any difficulties in applying the method to other vision problems, such as detection and segmentation? I do not mean to add experiments (though it would be great to do so), I just want to see the major difficulties of generalizing the method.

**Limitations:**

Please see the above comments.

**Strengths And Weaknesses:**

Strengths

1. The studied problem is important and interesting.

2. The proposed approach is sound and effective. The idea is reasonable and the experimental results are good.

3. The paper is well presented.

Weaknesses

I did not see significant weaknesses of this paper. Although the solution is still preliminary (in the context of the large CV community) and difficult to scale up to larger datasets, the proposed method is indeed a very good progress on this direction. There are some limitations that I hope the authors to address or explain, which I will write down in the following part.

---

> ### Author Response · Authors · 2022-08-02
> **Answer to Reviewer jSgu (2/2)**
>
> > Q3: If one has extracted, say, ten images per class and hopes to add five images per class for better performance, does the algorithm need to start from the very beginning (to extract 15 images per class), or can it start from the current point? In other words, what is the incremental performance of the proposed approach?
>
> It is an interesting question to see the incremental performance as it is essential for practitioners. However, we do not think our method has the desired adaptivity we hope. There are two ad hoc solutions. One is to ignore the previous distilled data and distill five images per class using a different random seed. However, since the target dataset is kept the same, it is expected to see that there will be a huge information overlap among the two sets of distilled images. Therefore, the improvement provided by the second distillation will be very marginal. The other solution is to consider the previously distilled images when we distill another set of images by including them in the meta-gradient computation. This way, we can minimize the information overlap between the two distillation phases. However, we do not expect to see any training efficiency improvement compared to distilling 15 images per class from scratch because the size of Gram matrices are the same, and it is likely to take a similar amount of steps for all synthetic images to converge.
>
> > Q4: Are there any difficulties applying the method to other vision problems, such as detection and segmentation? I do not mean to add experiments (though it would be great to do so). I just want to see the major difficulties of generalizing the method.
>
> Generalizing our method to other vision problems or to other domains like text or video seems to be an interesting future direction. We do not see significant difficulties when applying to other problems as long as we are in a supervised learning setting and we can formulate a regression problem. For example, for the image segmentation task where both the inputs and labels are images, we can still compute the Gram Matrix of inputs using the neural network feature and use KRR to compute the target labels (image) based on the training labels (images). We leave it for future work to explore this interesting direction.

---

> ### Author Response · Authors · 2022-08-02
> **Answer to Reviewer jSgu (1/2)**
>
> We thank you for your efforts in the review process; all questions are very insightful. If the following comments address your concerns, we would be grateful if you could consider increasing the review score.
>
> Three of your questions, Q1 (scale to high-resolution images), Q2 (scale to more distilled data), and Q3 (the adaptability of the algorithm), are closely related to each other and suggest a common optimization challenge when we scale up the algorithm. We believe that the current parameterization of the distilled data in raw pixel space is inefficient and not amicable to optimization. Therefore, we can do little to address those problems elegantly without significantly modifying the current algorithm. A better way to address those problems is to parameterize the distilled data using a neural network. In this way, 1) we can use all the tricks people developed in the past to train neural networks to ease the underlying optimization problem; 2) we can bake the inductive bias into the neural network design to improve the generation quality and use the parameter sharing to improve memory efficiency; 3) we can train an infinite data generator that tailored to the downstream task. We think this improvement is out of the scope of this submission, and we leave it for future work.
>
> > Q1: What is the major difficulty that prevents the proposed method from being applied to large images, e.g., 256x256 images? If one uses a small resolution setting to extract synthesized images (e.g., 64x64) and uses techniques like GAN to up-sample it larger (e.g., 128x128), does it bring good performance on large datasets (e.g., ImageNet)?
>
> Optimization is the main difficulty in scaling the algorithm to high-resolution images, and it becomes increasingly challenging to capture the correlation between pixels and different examples. It is a good idea to use the GAN approach to up-sample the data, but the main focus of our work is to propose a novel training objective. We follow the previous evaluation protocols and architectures closely. We will leave the generator approaches for future work.
>
> > Q2: What is the time and memory consumption for the ImageNet experiments, in particular, how long did the program run, and how many GPUs (or other devices) did it use? In addition, if one hopes to extract more training instances from ImageNet (e.g., 5, 10, 20, ... images per class), what is the computational cost w.r.t. the number of training instances?
>
> The largest experiment we have run was the ImageNet-1K (distilling two images per class, 2000 in total). This experiment was run on an A100 GPU with 40GB memory on AWS for a week (2 million gradient updates, including various checkpoints evaluation). The scalability with respect to the number of training instances is shown in Appendix C.5. We cannot extract more than 10K training instances from ImageNet due to our hardware constraints. Our current codebase does not support distributed training, so the number of distilled images we can extract is bounded by the GPU memory we can use. This engineering problem can be solved using the KIP paper's distributed kernel computation framework (https://arxiv.org/abs/2107.13034).
>
> Using all the synthetic data points to compute the meta-gradient is a limitation of the current method since the matrix inversion dominates the time complexity in Equation 2, which is O(N^3). It is acceptable with a few thousand images but becomes extremely expensive with tens of thousands of images. An ad hoc solution to distill more data points is to split the whole dataset into groups, either by class or randomly, and perform independent distillation like in Section 5.1. This is also the standard practice for previous methods. However, we observe a performance drop when we use such a strategy since the independent distillation may generate redundant information or fail to capture the distinguishable features that can only be identified considering the whole dataset. This problem is caused by not optimizing all parameters jointly, which can be potentially addressed by better parameterizing the distilled data. For example, we can parameterize the distilled data using a neural network. In that case, we can use a subset of synthetic data to compute the meta-gradient and optimize all model parameters jointly. This improvement is out of the scope of this submission, and we leave it for future work.

---

> ### Comment · Reviewer_JSgu · 2022-08-09
> **Post-rebuttal comments**
>
> I would like to thank the authors for the detailed responses.
>
> As I said in the initial review, there are no major weaknesses in this paper. Although the method does have some limitations (e.g. large computational costs -- I appreciate the authors for being honest and open to improvements), the method is already much better than prior ones, which deserves publication at NeurIPS.

---

### Official Review · Reviewer_p7tg · 2022-07-10

**Rating:** 7
**Confidence:** 4
**Soundness:** 3 good
**Presentation:** 3 good
**Contribution:** 3 good

**Summary:**

This paper proposes to use Kernel Ridge Regression on the last layer of a slowly updated feature extractor to perform Dataset Distillation. There are several key contributions in this paper, including demonstrating KRR can work well in DD, using model pools, fast training speed and state-of-the-art results on DD benchmarks. The experiment section shows results from both DD and other applications including continual learning and membership inference attacks.

**Questions:**

The paper nicely contributes to the DD task. I have two questions:
1. How does the algorithm compute the kernel matrix(see above)
2. Have the authors considered using implicit differentiation[24] instead, which also has low memory requirements? What would be the advantage of using KRR?

**Limitations:**

This work has potential benefits in improving privacy and understanding how neural networks perceive a task.

**Strengths And Weaknesses:**

Overall, this paper presents a practical method that performs well both in terms of performance and training time.

Strengths:
+ The paper contributed a new algorithm that uses KRR on the last layer and slowly updates the feature backbone to perform dataset distillation.
+ The paper well presents the algorithm and the related techniques and tricks for training
+ More stable performance on cross-architecture transfer is shown in table 2
+ The authors show that the algorithm can be used for higher-resolution training such as 128x128 images and 64x64 ImageNet dataset.

Weakness
- I'm curious how the authors handled the kernel computation in equation 2. If not mistaken, it seems the algorithm needs to compute the kernel for the full real dataset? Do you use minibatches instead?

Minor (no need to compare)
There are two recent new works on the same task, it might be worth adding them to the discussion/related work part.
1. Dataset Condensation via Efficient Synthetic-Data Parameterization
2. Remember the Past: Distilling Datasets into Addressable Memories for Neural Networks

---

> ### Author Response · Authors · 2022-08-02
> **Answer to Reviewer p7tg**
>
> We thank you for your efforts in the review process. The first question brings up an important potential confusion and the second question allows us to explain some hidden insights of our algorithm. If the following comments address your concerns, we would be grateful if you could consider increasing the review score.
>
> > Q1: I'm curious how the authors handled the kernel computation in equation 2. If not mistaken, it seems the algorithm needs to compute the kernel for the full real dataset. Do you use mini-batches instead?
>
> Thanks for pointing out this confusion. We can see that it comes from the fact that we introduced our algorithm using the full batch training notation in Equation 2, but Algorithm 1 (Line 3) samples a minibatch from the real labeled dataset. We have added a footnote after "d is the feature dimension" as follows: In practice, we use all the synthetic data and sample a minibatch from the real dataset to compute the meta-gradient (Algorithm 1).
>
> You may notice that the use of all the synthetic data can be a limitation when we want to distill more data rather than just a few thousand. An ad hoc solution is to split the whole dataset into groups, either by class or randomly, and perform independent distillation like what we do in Section 5.1. However, we observe a performance drop when we use such a strategy since the independent distillation may generate redundant information or fail to capture the distinguishable features that can only be identified considering the whole dataset. Essentially, this problem is caused by not optimizing all parameters jointly, which can be potentially addressed by a better parameterization of the distilled data. For example, we can parameterize the distilled data using a neural network. In that case, we can use a subset of synthetic data to compute the meta-gradient and optimize all model parameters jointly. This improvement is out of the scope of this submission, and we leave it for future work.
>
> > Q2: Have the authors considered using implicit differentiation [24] instead, which also has low memory requirements? What would be the advantages of using KRR?
>
> It is a great question. Here are some problems associated with implicit differentiation: 1) Implicit differentiation is based on the implicit function theorem (IFT), which assumes the inner optimization has converged. However, neural network optimization (the inner-level problem in our paper) is typically far from convergence (https://arxiv.org/pdf/2110.06256.pdf). Previous approaches tend to neglect this underlying assumption and apply implicit differentiation. Although IFT can work in practice, we argue that our proposed approach provides a better approximation of the actual neural network optimization dynamics, as demonstrated by our empirical experiments. (IFT results Figure 6 in http://proceedings.mlr.press/v108/lorraine20a/lorraine20a.pdf) 2) The meta-gradient computed by implicit differentiation also assumes the uniqueness of the inner solution and certain regularity conditions. However, the inner optimization in dataset distillation is typically overparameterized (deep neural network), so the set of optima forms a non-trivial manifold, which does not satisfy the IFT's assumptions. 3) In implicit differentiation, we need to compute the inverse Hessian, which is typically approximated by an iterative linear solver, such as truncated gradient or Neumann series. Though they have low memory requirements, they are not very time-efficient. In practice, we need to tune an extra hyperparameter - truncation step K to achieve the performance and time trade-off we want. 4) Implicit differentiation is computed at some approximated solution and does not care about how we arrive at this point. However, the training dynamics of neural networks are complicated and affect the solution we find. Therefore, we may not want to ignore this dependency. For more details regarding bi-level optimization, here are two interesting papers: https://proceedings.mlr.press/v162/vicol22a/vicol22a.pdf, https://www.cs.toronto.edu/~lorraine/papers/implicitReg.pdf.
>
> Compared to implicit differentiation, our methods have the following advantages: 1) Our method is based on unrolled optimization, and we do not make any assumptions as in the implicit function theorem. The method is well-defined regardless of how many steps we unroll. The problem with the unrolled optimization is the approximation error or so-called truncation bias. We alleviate this problem by training a subset (last layer) of the neural network to convergence. 2) Our method considers the training dynamics of neural networks by model online update. Besides, we decouple the meta-gradient computation from the choice of inner loop optimizer. Thus, we can use any optimizer to improve the optimization or generalization of the inner loop without worrying about overfitting the inner loop learning algorithm (e.g., number of training iterations).

---

> ### Comment · Reviewer_p7tg · 2022-08-07
> **Thank you for the responses, several additional questions**
>
> Thank you for the responses and clarifications. I have several additional questions regarding this method:
>
> 1. It looks like this paper uses the KRR objective function used/proposed in Nguyen et al. ICLR 2021[22]? If so, it might be good to explicitly refer to that paper in the objective function paragraph and make it clear that they did use/propose it first. Readers can also easily trace back and see the progress along this line of work.
>
> 2. What would be the main modification to make the original KRR work? Is it you're using a better model pool or models with higher feature dimensions or moving backbones?
>
> 3. Did the authors try to retrieve the most similar images in the real dataset to make sure it's not actually selecting/copying key exemplars from the dataset?
>
> 4. It looks like part of the model's intuition is: I'd like to produce features (with distilled data) such that the solved weights can classify real images correctly. Is this almost quite similar to feature matching, but instead of directly matching them using standard metrics, you matched it with an optimization criterion? Do you think this is the reason that your algorithm produces quite realistic-looking images?

---

> > ### Author Response · Authors · 2022-08-08
> > **Response to Reviewer p7tg (2/2)**
> >
> > > Q3: Did the authors try to retrieve the most similar images in the real dataset to make sure it's not actually selecting/copying key exemplars from the dataset?
> >
> > It is a great question, which is why we provide four videos to visualize the distillation process in the Appendix. The distilled data is the product of the optimization procedure, and it converges to a similar point whether it is initialized from the real image or random noise. As you can see in the videos, the distilled data is far away from its initialization. Intuitively, the distill data can be considered the "principle component" of a dataset. Instead of selecting or copying the exact images from the dataset, our method synthesizes the images that best reflect the most representative pattern in the original dataset.
> >
> > Another evidence that "learning" is happening is that a pair of distilled images and labels are highly correlated. Suppose you interpolate the label value between two different classes and only learn the image. You can see the learned image interpolates between the two classes as well, which suggests our method tries to synthesize the best image to reflect its label.
> >
> >
> > > Q4: It looks like part of the model's intuition is: I'd like to produce features (with distilled data) such that the solved weights can classify real images correctly. Is this almost quite similar to feature matching, but instead of directly matching them using standard metrics, you matched it with an optimization criterion? Do you think this is the reason that your algorithm produces quite realistic-looking images?
> >
> > It is an interesting question. Intuitively, our method predicts by computing the similarity (measured by the conjugate kernel) between a real image and a set of distilled images. If we want this similarity measure to be accurate, making the distilled data features look closer to that of real data is better. Therefore, realistic-looking seems like a natural result of applying our method. However, multiple factors can affect the appearance of our distilled images. For example, Appendix C.4 shows that learning the label can play an important role; Appendix C.6 shows that the width, depth, normalization, and skip connection can also affect the appearance. Therefore, we also need to choose the right model architecture to generate realistic-looking images.
> >
> > However, we do not think our method is very similar to feature matching. The key difference is that our objective is indeed task-driven and requires labels to find the most discriminative feature for a particular task. However, the feature matching generally matches some feature statistics of the real and distilled data. It is likely that they only capture some general features that are useful to recover the data but not very useful for the downstream task. This is why the previous methods, like gradient matching and distribution matching, fail on the fine-grained classification task (Table 1. CUB-200), as many fine-grained classes share the same distilled features, thus providing no discriminative ability. Another difference is that feature matching objectives are not amenable for label learning, which is crucial for complex label spaces (e.g., ImageNet-1K, CUB-200). Thus, it is difficult for them to consider the class similarity information, so it becomes hard to learn the sharable information across different classes, resulting in a poor compression rate. Moreover, another drawback of surrogate objectives like gradient matching or feature matching is that it is unclear how far it is from the true objective. It is pretty tricky to measure how "biased" those objectives are. It is unclear whether the correct set of features (e.g., last layer or middle layer's activation or gradient) is used or whether a proper distance measure (e.g., L2 distance, cosine distance) is used. We conjecture that this biased objective is why previous methods do not show real-looking images.
> >
> > > Q5: Related work
> >
> > Thanks for pointing out the related works [1][2]. They are both recent works in dataset distillation but are not so closely related to ours. Our work mainly focuses on the training objective, while the two related works focus on the parameterization of the distilled data, which are orthogonal to ours. But, we are happy to discuss them when we have more space.
> >
> > [1] Dataset Condensation via Efficient Synthetic-Data Parameterization
> > [2] Remember the Past: Distilling Datasets into Addressable Memories for Neural Networks

---

> > > ### Comment · Reviewer_p7tg · 2022-08-09
> > > **Thank you for the response**
> > >
> > > Thank you for the explanations and responses. This paper is a great work on making KIP and bi-level optimization algorithms practical, and should be known by the field.

---

> > ### Author Response · Authors · 2022-08-08
> > **Response to Reviewer p7tg (1/2)**
> >
> > Thanks for the follow-up. It is our pleasure to answer your questions as all of them are very insightful and helpful in understanding our method better. Let us know if you have any further questions.
> >
> > > Q1: It looks like this paper uses the KRR objective function used/proposed in Nguyen et al. ICLR 2021[22]? If so, it might be good to explicitly refer to that paper in the objective function paragraph and make it clear that they did use/propose it first. Readers can also easily trace back and see the progress along this line of work.
> >
> > Sure, we modify the main text as you suggest (Line 109-110).
> >
> > > Q2: What would be the main modification to make the original KRR work? Is it you're using a better model pool or models with higher feature dimensions or moving backbones?
> >
> > We think the idea of KIP is excellent. The spirit of our method and KIP are similar as we both want to find a good approximation to the inner loop optimization. If you think more along this line and are familiar with NTK theory, you would likely come up with a similar idea. Indeed, KIP works well in certain cases (e.g., given enough computation resources, low-resolution images).
> >
> > However, KIP has several drawbacks that make it less practical. We show them in the decreasing order of significance below.
> > - High computation cost: It requires thousands of GPU hours to perform a single experiment on low-resolution image datasets like CIFAR10. You can see more details in Appendix B of https://arxiv.org/abs/2107.13034, and you can also learn more from the neural tangent library https://github.com/google/neural-tangents#cnn-with-pooling).
> > - Poor Scalability: KIP can not scale well to high-resolution images (time complexity is O(d^2) if using convolutional neural networks with pooling layers, where d is the number of pixels) or many data points.
> > - Discrepancy between finite-width neural networks: KIP uses analytical NTK to compute the meta-gradient, which assumes the neural network to be infinitely wide and operates in the kernel regime. However, a standard finite-width neural network does not satisfy these assumptions, and its training dynamic is quite complex. Training a finite-width neural network on the distilled data causes a significant performance drop in evaluation.
> > - Architecture Constraints: Many modern architectures with a normalization layer (e.g., BatchNorm) do not have an analytical NTK expression. So, the set of model architectures that KIP can use is quite limited.
> >
> > There are three main components in our method:
> > - (1) Conjugate kernel using a wider neural network (higher feature dimension)
> > - (2) Online model update (moving backbones)
> > - (3) Model pool
> >
> > We think all of them are important, but in decreasing order of significance. We also provide additional tricks to further improve the performance in Appendix D.
> >
> > The first component is the most crucial one, designed to approximate the inner loop optimization efficiently. Compared to NTK approximation in KIP, our method is at least four orders of magnitude faster, which makes our method practical. You can find more ablation studies regarding the significance of the architecture (e.g., width, depth, and normalization) in Appendix C.6. Note that KIP uses the infinite-wide neural network to compute the meta-gradient for the distilled data, which is indeed using a much wider neural network than ours.
> >
> > The second component (online model update) aims to take the neural network training dynamic into account, which can improve the performance on CIFAR100 1 Img/Cls by 2% (Appendix Figure 9(b)). Essentially, this component is designed to account for errors caused by linear approximation. We want to take the complex NN training dynamics into account and gather gradient information from all stages of training rather than just the initialization. (1) + (2) form the analogy of 1-step TBPTT. People generally think 1-step TBPTT is bad as it causes significant truncation errors. However, the kernel approximation elegantly addresses this problem as it is equivalent to training the last layer into convergence.
> >
> > The third component is a generalization of the iterative model reinitialization techniques, which can improve the performance on CIFAR100 1 Img/Cls by 1% (Appendix Figure 9(a)). A direct motivation of this design is observing the cyclic behavior of loss when using an iterative model reinitialization. It suggests that the iterative model reinitialization scheme wastes some computation when it overfits a particular training trajectory. Besides, from the perspective of meta-learning, it is always good to have a diverse meta-train task to learn a more generalized meta-parameter. Therefore, we use the "model pool" idea to provide diverse meta-training tasks.
> >
> > To sum up, our method shares the same spirit as KIP and effectively addresses the drawbacks of KIP.

---

> > ### Author Response · Authors · 2022-08-09
> > **Let us know if you have any other questions**
> >
> > Dear reviewer,
> >
> > Thanks for your efforts in the reviewing process! Let us know if you have any further questions before the end of the author-reviewer discussion phase.
> >
> > Thanks,
> > Authors

---

### Official Review · Reviewer_aP6S · 2022-07-11

**Rating:** 8
**Confidence:** 5
**Soundness:** 4 excellent
**Presentation:** 4 excellent
**Contribution:** 4 excellent

**Summary:**

This paper proposes a new method of dataset distillation  called “Neural Feature Regression with Pooling.” This work avoids using a surrogate objective  like previous works (DSAm DM, MTT) while also circumventing the computational constraints of other methods that optimize the true objective (DD, KIP). Like other methods, FRePo trains teacher networks from which to obtain “meta-gradients.” Unlike other methods, FRePo obtains the meta gradient by finding the closed form solution of the Kernel Ridge Regression problem posed by the final layer (linear classifier) of the teacher network. This method shows state-of-the-art results, outperforming previous methods in nearly every setting. Two other applications in continual learning and membership inference defense are also explored.


**Questions:**

While my questions should be easy to answer, I think they are quite critical to the validity of this work. I will happily adjust my rating if they are addressed.

1. How does FRePo perform on the original architecture? (I believe you call it DC-Conv in the appendix). Even if these results are significantly lower, they should be included. They can even be left to the appendix as long as it is made VERY clear in the body that a new, wider architecture is being used and the reasons for this change are properly explained. (You mentioned in the appendix that the reason is to ensure the KRR behaves well. This explanation makes sense, but it is still a limitation of this work.)

2. Are the re-evaluations in Table 1 done using the \emph{exact} same architecture used to get the FRePo numbers? (i.e., is the same normalization type being used?) If not, how do the other algorithms perform using BatchNorm.

3. Regardless of the answer to the previous question, how does FRePo perform using InstanceNorm?

4. How is FRePo’s cross-architecture performance when trained on Conv-IN like all the other methods in Table 2 (aside from KIP)?

5. If FRePo is faster and uses less memory than MTT, DSA, and DM, why were results for CIFAR100 and T-ImageNet at 50 img/cls not included here when they were included in previous works?

-----

Edit: All my questions and concerns have been addressed.

**Limitations:**

The limitations on the size of the teacher model seem unique to this method among dataset distillation works and should be clearly addressed.

**Strengths And Weaknesses:**

Strengths

The paper is very well written with very few grammatical errors (e.g., subject-verb agreement). All figures and tables are formatted well and clearly communicate the authors’ ideas.

As for the method itself, FRePo clearly outperforms the other methods in the explored settings. Another dataset (ImageNet1k resized) is also introduced and will likely be a new evaluation metric for future dataset distillation works.

Extensive visualizations are also included in the appendix.

--------

Weaknesses

I was disappointed to see that the fact that a new model was used (with respect to previous dataset distillation works) was not mentioned until deep into the appendix. While advocating for the adoption of a new backbone model is fine, this should be made very clear in the body of the paper. I understand that the authors re-evaluated the previous methods using this new architecture in Table 1, but the fact that a new architecture was used was not clear at all from the body alone. The authors should have also included results from their method on the original architecture. If this yields poor results, then this is a significant limitation of the method and should be addressed as such (but by no means detracts from the merit of this new contribution).

Furthermore, even given the appendix, it still remains unclear if these re-evaluations of previous methods use an architecture identical to the new one used by FRePo. The methods should have been re-evaluated with batch-norm, and results for FRePo using instance norm should have also been included. It is unclear how much of the improvement over previous state of the art is due to the algorithm, normalization type, or number of channels.

Similarly, Table 2 should also \textit{at least} include an additional row for FRePo trained using Conv-IN.

Also, if FRePo is faster and uses less memory than MTT, DSA, and DM, why were results for CIFAR100 and T-ImageNet not included here when they were included in previous works?

One other tiny thing: your labels for “Samoyed” and “Golden Retriever” seem to be swapped in all of your ImageWoof figures :P

---

> ### Author Response · Authors · 2022-08-02
> **Answer to Reviewer aP6S (2/2)**
>
> > Q3: Regardless of the answer to the previous question, how does FRePo perform using InstanceNorm?
>
> See Appendix C.6 Table 16 and Table 17. We observe that instance normalization performs slightly worse than the default batch normalization. However, Table 15 suggests that the drawback of the instance norm is the transferability. The distilled data trained using instance normalization transfer less well to other architecture, especially those without normalization.
>
> > Q4: How is FRePo's cross-architecture performance when trained on Conv-IN like all the other methods in Table 2 (aside from KIP)
>
> As shown in the Table below or Table 15 in Appendix C.6, the distilled data trained with Conv-IN transfers less well to architectures than the distilled data trained by Conv-BN. However, the distilled data generated by FRePo (Conv-IN or Conv-BN) still outperforms the previous methods on ResNet, VGG, and AlexNet.
>
> | | | Conv | Conv-NN | ResNet-DN | ResNet-BN | VGG-BN | AlexNet |
> | :---: | :---: | :---: | :---: | :---: | :---: | :---: | :---: |
> | DSA | Conv-IN | 53.2+-0.8 | 36.4+-1.5 | 42.1+-0.7 | 34.1+-1.4 | 46.3+-1.3 | 34.0+-2.3 |
> | DM | Conv-IN | 49.2+-0.8 | 35.2+-0.5 | 36.8+-1.2 | 35.5+-1.3 | 41.2+-1.8 | 34.9+-1.1 |
> | MTT | Conv-IN | 64.4+-0.9 | 41.6+-1.3 | 49.2+-1.1 | 42.9+-1.5 | 46.6+-2.0 | 34.2+-2.6 |
> | KIP | Conv-NTK | 62.7+-0.3 | 58.2+-0.4 | 49.0+-1.2 | 45.8+-1.4 | 30.1+-1.5 | 57.2+-0.4 |
> | FRePo | Conv-IN | 59.2+-0.3 | 56.2+-0.2 | 51.1+-0.8 | 50.8+-0.2 | 51.8+-0.3 | 55.3+-0.8 |
> | FRePo | Conv-BN | 65.5+-0.4 | 65.5+-0.4 | 58.1+-0.6 | 57.7+-0.7 | 59.4+-0.7 | 61.9+-0.7 |
>
>
> > Q5: If FRePo is faster and uses less memory than MTT, DSA, and DM, why were results for CIFAR100 and T-ImageNet at 50 img/cls not included here when they were included in previous works?
>
> |Img/Cls|DSA|DM|MTT|FRePo|
> |--|--|--|--|--|
> |50| 42.8$\pm$0.4|43.6$\pm$0.4| 47.7$\pm$0.2 |44.3$\pm$0.2|
>
> We provide the CIFAR100 with 50 img/cls result in the table above. Our method performs better than DSA and DM but performs worse than MTT. There are several potential reasons for this observation: 1) Since we use the same hyperparameter setting as all other experiments to make our experiments consistent, the current hyperparameter is suboptimal for this setting (CIFAR100 with 50 img/cls). For example, the maximum number of online model updates (100) is too small to distill 5000 distilled images. Setting it to a larger value would yield better performance when we distill more data (Appendix D). Note that all previous methods tune the hyperparameter when they switch to different datasets and distill different numbers of images 2) The KRR is less stable when the feature dimension is closed to the data dimension. Tuning the kernel ridge regularizer could improve the performance. 3) There is a more complex optimization problem. It is difficult to capture the correlation between pixel and 5000 distilled images. We leave it for future work to improve upon the current method.
>
> We are not able to run Tiny-ImageNet due to our hardware constraints. Our current codebase does not support distributed training, so the number of distilled images we can extract is bounded by the GPU memory we can use. Note that MTT requires 6 RTX6000 GPU with 144GB to run this experiment, while all our experiments can be done in a single GPU. This engineering problem can be solved using the KIP paper's distributed kernel computation framework (https://arxiv.org/abs/2107.13034). But since it is not open-source, we are still working in progress to extend our codebase to support multi-GPU training.
>
> > Q6: Role of normalization type and the number of channels.
>
> We conduct several ablation studies on CIFAR100 with 1 Img/Cls to study the role of the normalization layer and the role of width. Tables 12-14 and Figure 15-16 show that 1) Training with Conv-BN and evaluating using Conv-NN yields the best performance, which is our default choice; 2) Training with Conv-BN yields the most generalizable and transferable images as it performs well for other architectures; 3) Evaluating using Conv-GN seems to be the best choice if the training architecture is unknown. 4) Evaluating using Conv-NN is an excellent way to see whether the inductive bias of architecture has been distilled to the dataset since it is very sensitive to the training architecture. There, it is not a good idea to use this architecture when the training architecture is unknown; 5) Training the distilled data using a wider network Conv achieves slightly better performance than a narrower network DCConv. 6) Visually, we find that Conv does generate better images than DCConv and BN, IN and GN do not affect the performance too much when Conv is used. 7) Though using DCConv-IN is not the best choice for our method, we still outperform the previous methods. We achieved an accuracy of 24.8, compared to DSA (16.8), DM(12.2), KIP(15.7), and MTT (24.3)

---

> ### Author Response · Authors · 2022-08-02
> **Answer to Reviewer aP6S (1/2)**
>
> We thank you for your efforts in reviewing and looking into every detail of our work; all questions are very useful. If the following comments address your concerns, we would be grateful if you could consider increasing the review score.
>
> We are sorry to hear that you feel disappointed about the architectural change. Due to space, all the model changes, data preprocessing, and other implementation details were only present in Appendix A of the initial submission.  We have updated Section 4.1 to reflect your suggestions in the current version. We have also included Table 16 for a comparison between the original and the new architectures. Here, we provide a summary of the architectural choices. In all the baseline experiments reported, we found methods such as DM and MTT consistently favour the original DCConv-IN architecture. They tend to achieve 1-5% lower accuracy under our modified architecture, while FRePo achieves higher accuracy using Conv-BN. We provide a short summary of MNIST and CIFAR10 below. FRePo using Conv-BN achieves better performance with much faster convergence on a single GPU. For a more comprehensive comparison of other datasets and experimental details, see Appendix C.6.
>
> |         |      | DSA | DM  | MTT  |              Conv-BN              |              Conv-IN             |    DCConv-IN   |
> |:-------:|:----:|:-----------------------------:|:-------------------------------:|:---------------------------------:|:---------------------------------:|:--------------------------------:|:--------------:|
> |  MNIST  |   1  |         $88.7 \pm 0.6$        |          $89.9 \pm 0.8$         |           $91.4 \pm 0.9$          |  $\textbf{93.0} \pm \textbf{0.4}$ |          $92.9 \pm 0.5$          | $92.4 \pm 0.5$ |
> |         |  10  |         $97.9 \pm 0.1$        |          $97.6 \pm 0.1$         |           $97.3 \pm 0.1$          | $\textbf{98.6} \pm \textbf{0.1}$ |          $98.9 \pm 0.1$          | $98.4 \pm 0.1$ |
> |         |  50  |         $99.2 \pm 0.1$        |          $98.6 \pm 0.1$         |           $98.5\pm 0.1$           |           $99.2 \pm 0.0$          | $\textbf{99.4} \pm \textbf{0.1}$ | $98.8 \pm 0.1$ |
> | CIFAR10 |   1  |         $36.7 \pm 0.8$        |          $31.0 \pm 0.6$         |          $46.3\pm 0.8$          | $\textbf{46.8} \pm \textbf{0.7}$ |          $45.1 \pm 0.5$          | $41.3 \pm 0.5$ |
> |         |  10  |         $53.2 \pm 0.8$        |          $49.2 \pm 0.8$         |          $ 65.3 \pm 0.7$         |  $\textbf{65.5}\pm \textbf{0.4}$  |          $59.1 \pm 0.3$          | $59.6 \pm 0.3$ |
> |         |  50  |         $66.8 \pm 0.4$        |          $63.7 \pm 0.5$         |          $ 71.6\pm 0.2 $          |  $\textbf{71.7}\pm \textbf{0.2}$  |          $69.6 \pm 0.4$          | $63.6 \pm 0.2$ |
>
> Here is a summary of the new experiments:
> -  Add experimental results similar to Table 1 using Conv-IN (Q3) and DCConv-IN (Q1) to Appendix C.6 Tables 16 and 17.
> - Add cross-architecture transfer performance using Conv-IN (Q4) to Appendix C.6. Table 15.
> - Add an ablation study on normalization and model width on CIFAR100 with 1 Img/Cls to Appendix C.6.  Table 12-14.
>
> > Q1: How does FRePo perform on the original architecture? (I believe you call it DC-Conv in the appendix). Even if these results are significantly lower, they should be included. They can even be left to the appendix as long as it is made VERY clear in the body that a new, wider architecture is being used, and the reasons for this change are properly explained. (You mentioned in the appendix that the reason is to ensure the KRR behaves well. This explanation makes sense, but it is still a limitation of this work.)
>
> See Appendix C.6 Table 16 and Table 17. We observe that DCConv works reasonably well when distilling a small number of images (~100). The performance degrades a lot when distilling 1000 images from CIFAR100 because the KRR component needs a larger feature dimension to perform well when we distill more data.
>
> > Q2: Are the re-evaluations in Table 1 done using the emph{exact} same architecture used to get the FRePo numbers? (i.e., is the same normalization type being used?) If not, how do the other algorithms perform using BatchNorm?
>
> Yes, we indeed made an optimistic estimation of the previous methods. As we mention in Appendix A.1, we run four settings for previous methods, namely {original data preprocessing, FRePo ZCA processing} x {DCConv, FRePo Conv (wider, BatchNorm)} and we pick the setting that turns out to be the best for the previous methods. Moreover, we report the original paper's performance if it is better than our reproducing results. In our experiments, we observed that FRePo ZCA processing could yield better performance, but FRePo Conv does not seem to help and yields a worse performance.

---

> > ### Comment · Reviewer_aP6S · 2022-08-04
> > **Response to Rebuttal**
> >
> > Thanks so much for your detailed response. Most of my concerns have been addressed.
> >
> > However, there are still a few things that I think need to be better addressed in the main text.
> >
> > I think this is really great work, but these last few concerns hold it back.
> >
> > Please submit another revision when these concerns have been addressed.
> >
> > -----
> >
> > 1.
> >
> > Based on Table 16, KIP and/or MTT out-performs FRePo in every CIFAR-10 and CIFAR-100 setting (except CIFAR-100, 1ipc where MTT and FRePo are evenly matched). I assume MTT would also out-perform FRePo on the omitted CIFAR-100, 50 ipc setting based on the table you included here.
> >
> > So essentially, FRePo **only** achieves state-of-the-art performance when this new larger architecture is used.
> >
> > Note that I do not think that this detracts from the work **at all**. On the contrary, I think the whole dataset distillation community should start using larger, less toy-ish networks as benchmarks.
> >
> > In general, I also do not think that a new method *needs* to show state-of-the-art results in every setting to be a significant contribution to the community.
> >
> > However, I *do* think that this needs to be **much** more clearly addressed in the main text. It would be enough to simply add to 4.1 something like: "We use a larger architecture than previous dataset distillation works because the KRR problem is not well-behaved when the feature dimension is low, causing our methods performance to drop. Results on the original architecture are included in Appendix C.6."
> >
> > -----
> >
> > 2. I appreciate you showing the CIFAR-100 50 ipc results here, but please also include them in the main text (Table 1). There is nothing wrong with not having the best numbers in every last setting.
> >
> > -----
> >
> > 3.
> >
> > Please, please, please fix your ImageWoof labels.
> >
> > On further inspection, all of them (except Dingo by chance) actually seem to be incorrect.
> >
> > It looks like you copied the class names from here (https://github.com/fastai/imagenette), but this is not at all the order in which you are displaying them.
> >
> > Based on Figure 30 where you display all the classes, it looks like the order you are using is:
> >
> > Border Terrier, English Foxhound, Golden Retriever, Rhodesian Ridgeback, Old English Sheepdog, Australian Terrier, Beagle, Dingo, Samoyed, Shih-Tzu

---

> > > ### Author Response · Authors · 2022-08-05
> > > **Response to Reviewer aP6S**
> > >
> > > Thanks for the follow-up. We have fixed your concerns as below, and you can also see the new changes in the updated main text and appendix. Let us know if you have any further questions.
> > >
> > > Q1: Yes, we agree that under our current hyperparameter setting, FRePo only achieves state-of-the-art performance when the new wider architecture is used. We have updated Section 4.1 as you suggest. We agree with the point that “the whole dataset distillation community should start using larger, less toy-ish networks as benchmarks.” Other than model architecture, we think the data preprocessing, data augmentation, model initialization, and model training should all be unified. We can see some challenges in this line of work. For example, suppose we want to use the same architecture for both training and evaluation for all methods (current practice). In that case, it may not be very fair as different algorithms seem to prefer different settings. However, suppose we only care about a generalized distilled dataset that can transfer well to a wide range of architectures (cross-architecture transfer experiment). In that case, evaluating the data quality will be much easier as we can use the same evaluation protocol for all. The training setups, including the model architecture, are just hyperparameters of a particular algorithm.
> > >
> > > Q2: Sure. We have added the result to the main text (Table 1) and Appendix (Table 7, 16, and 17).
> > >
> > > Q3: Thanks for pointing out the correct labels. We have fixed them (Figure 1c and Figure 30).

---

> > > > ### Comment · Reviewer_aP6S · 2022-08-05
> > > > **Thanks!**
> > > >
> > > > Thanks for addressing all this!
> > > >
> > > > To the end of standardizing dataset distillation, there is a new benchmarking framework out in the dataset track: https://openreview.net/forum?id=Bs8iFQ7AM6
> > > >
> > > > It seems like a good start towards standardizing evaluations.
> > > >
> > > > Anyways, great work!

---

### Official Review · Reviewer_nCf8 · 2022-07-11

**Rating:** 7
**Confidence:** 2
**Soundness:** 3 good
**Presentation:** 3 good
**Contribution:** 3 good

**Summary:**

This paper proposes a new data distillation approach based on neural feature regression that is similar to a truncated backgprop through time using a pool of models. The approach sets a new state-of-the-art results both in terms of accuracy and training efficiency.  Ample experiments showcase the advantages of the proposed approach.

**Questions:**

Please see the Strengths and Weaknesses section

**Limitations:**

It would be helpful if the authors would have specified in the form where each content is addressed instead of simply stating Yes/No.

**Strengths And Weaknesses:**

Strengths:
- State-of-the-art results while being significantly more efficient
- Well written and structured with in depth ablation studies

Weaknesses & Q:
- L173, why no augmentation is applied during training? Shouldn't this prevent overfitting too?
- How does this approach fair in comparison with state of the art when combined with few shot learning approaches (that were targeted for real data)?
- Since the paper mentions the computational advantages, perhaps this information could be added in the tables too. Thought figure 3 addresses this in part.
- Given the difficulty on modeling similar classes, it would be interesting to see how such method fair on fine grained classification (e.g. on CUB-200)

---

> ### Author Response · Authors · 2022-08-02
> **Answer to Reviewer nCf8**
>
> Thank you for your feedback in the review process. We appreciate all the insightful questions, especially the suggestion on the fine-grained classification experiment on CUB-200. If the following comments address your concerns, we would be grateful if you could consider increasing the review score.
>
> > Q1: Given the difficulty of modeling similar classes, it would be interesting to see how such a method fair on fine-grained classification (e.g., on CUB-200)
>
> We have evaluated our method on CUB-200-2011 (rescale to 32x32) and added the result to the main text (See the visualization in Appendix E.2). As shown in the table below, our method achieves significantly better performance than all other methods and real data, especially when we distill only one image per class. We observe that methods like DSA and DM that work reasonably well for coarse-grained image classification tasks fail to outperform the real data baseline on the fined-grained classification task. This is because DSA and DM learn the distilled data independently and fail to capture the similarity among different classes. In contrast, our method works well as we take into account the class similarity by considering all classes together and learning the label. The learned label also plays an important role at test time. Similar to the teacher label in knowledge distillation (https://arxiv.org/abs/1503.02531), our distilled label is soft and not only contains knowledge for the most likely class but also contains “dark knowledge” like class similarity for all other classes. As a result, training a model from scratch on our distilled data achieves much better performance than the previous methods that do not learn the label.
>
> Table: Test accuracies of models trained on the distilled data from scratch on CUB-200-2011 (http://www.vision.caltech.edu/datasets/cub_200_2011/), a fine-grained image classification dataset. Test accuracy on the full dataset is $21.74\pm0.64$.
>
> |Img/Cls|Real|DSA|DM|MTT|FRePo|
> |--|--|--|--|--|--|
> |1|1.43$\pm$0.11|1.29$\pm$0.09|1.61$\pm$0.06|2.16$\pm$0.05| 12.41$\pm$0.20|
> |10|5.36$\pm$0.31| 4.54$\pm$0.26 |4.38$\pm$0.16| OOM |16.84$\pm$0.12|
>
> > Q2: L173, why is no augmentation applied during training? Shouldn't this prevent overfitting too?
>
> Yes, we agree that data augmentation during training can alleviate the overfitting problem, but finding the correct data augmentation can be complex. We not only need to consider the data augmentation for two different stages (i.e., meta-gradient computation and online model update), but we also need to consider the data augmentation for two different data types (i.e., distilled data and real data). In our experiments, the same kind of data augmentation, cutout (https://arxiv.org/abs/1708.04552), for example, can improve the performance if applied to the online model update but hurt the performance if it is applied to the meta-gradient computation. Besides, different datasets may require different data augmentations. For example, we may want to use image flipping for datasets involving natural images but not for datasets containing digits like MNIST. Moreover, the optimal hyperparameters for different transformations (e.g., color jittering and scaling) are different and need to be tuned separately for each dataset.
>
> In our experiments, we found that applying the correct data augmentation could improve the final test performance by around 1–3%, especially when we distilled more data points. In Appendix D, we discuss the training time data augmentation and several other tricks that can improve the model's performance but are not included in the current method.
>
> > Q3: How does this approach fair in comparison with the state-of-the-art when combined with the few-shot learning approaches (that were targeted for real data)?
>
> We think it is interesting to see whether we can combine the dataset distillation techniques with few-shot learning approaches since the distilled data can be seen as a good prototype candidate (https://arxiv.org/pdf/1703.05175.pdf). However, exploring this idea is out of the scope of the current project since dataset distillation and few-shot learning are very different. Dataset distillation aims to learn a highly informative and condensed dataset, while few-shot learning focuses on the adaptability of the model to different learning scenarios. At test time, dataset distillation trains a model from scratch on the distilled data, while few-shot learning relies on the model trained during "meta-training." We leave it to future work to explore the combination of dataset distillation and few-shot learning.
>
> > Q4: Since the paper mentions the computational advantages, perhaps this information could be added to the tables too. Though figure 3 addresses this in part.
>
> Thanks for the suggestion. We have added four tables to the appendix (Appendix.C.5 Table 8-11), which include time per step and peak GPU memory usage for various model sizes and the number of distilled data points.

---

> > ### Comment · Reviewer_nCf8 · 2022-08-09
> > **Concerns largely addressed**
> >
> > Thank you for your reply, my concerns were largely addressed. Accordingly, I have updated my rating from 6 to 7.

---

### Author Response · Authors · 2022-08-02
**Author response - Summary**

We would like to thank all the reviewers for their time and effort in the review process. Overall, we are pleased to see that all reviewers recognize the key contributions of this work and recommend acceptance at NeurIPS. We summarize the discussion among each reviewer and the new experiments below.

### Clarification and Discussion
- Clarify the data augmentation. (@Reviewer nCf8)
- Clarify the meta-gradient computation. (@Reviewer p7tg)
- Clarify the evaluation protocol. (@Reviewer aP6S)
- Clarify the model architecture. (@Reviewer aP6S)
- Fix the ImageWoof labels. (@Reviewer aP6S)
- Discuss the drawbacks of implicit differentiation and the merits of FRePo in the context of dataset distillation. (@Reviewer p7tg)
- Discuss the drawbacks of KIP and our improvements to it. (@Reviewer p7tg)
- Discuss the drawbacks of feature matching objectives and how FRePo differs from them. (@Reviewer p7tg)
- Discuss why and how to make our distilled images look real. (@Reviewer p7tg)
- Discuss what the distilled data learn. (@Reviewer p7tg)
- Discuss several limitations and future work of FRePo (@Reviewer jSgu)

### Experiments
- Add table summaries for computational costs (similar to Figure 3) to Appendix C.5. (@Reviewer nCf8)
- Evaluate the proposed method (FRePo) on a fine-grained image classification dataset (Table 1) (CUB-200-2011 (http://www.vision.caltech.edu/datasets/cub_200_2011/)) (@Reviewer nCf8)
- Add CUB-200 visualizations to Appendix E.2 (@Reviewer nCf8).
- Add experimental results similar to Table 1 using Conv-IN and DCConv-IN to Appendix C.6. (@Reviewer aP6S)
- Add cross-architecture transfer performance using Conv-IN to Appendix C.6. (@Reviewer aP6S)
- Add an ablation study on normalization and model width to Appendix C.6. (@Reviewer aP6S)

---

### Meta-Review · Area_Chair_hxrW · 2022-08-28

**Recommendation:** Accept
**Confidence:** Certain

**Metareview:**

The paper proposes a new algorithm for dataset distillation, based on two key ideas:  (1) train a linear layer given the fixed feature extractor, and (2) use a diverse set of modes as feature extractors. The paper has received overwhelmingly positive reviews. Many reviewers find the algorithm effective, the paper well-written, and the results compelling. The rebuttal further addressed the concerns regarding the backbone models and missing experiments as well as provided additional clarifications. The AC agreed with the reviewers’ consensus and recommended accepting the paper.

**Award:**

Yes

---

### Decision · Program_Chairs · 2022-09-14

Accept